# Tip carbon encapsulation customizes cationic enrichment and valence stabilization for low K⁺ acidic CO₂ electroreduction

Zhitong Wang[1,2,7], Dongyu Liu [3,7], Chenfeng Xia[2,7] ✉, Xiaodong Shi [1,7], Yansong Zhou[4], Qiuwen Liu[5], Jiangtao Huang[5], Haiyan Wu[1], Deyu Zhu[2], Shuyu Zhang[4], Jing Li[1], Peilin Deng [1], Andrey S. Vasenko [3,6], Bao Yu Xia [2] ✉ & Xinlong Tian [1] ✉

Acidic electrochemical CO₂ conversion is a promising alternative to overcome the low CO₂ utilization. However, over-reliance on highly concentrated K⁺ to inhibit the hydrogen evolution reaction also causes (bi)carbonate precipitation to interfere with catalytic performance. In this work, under the screening and guidance of computational simulations, we present a carbon coated tip-like In₂O₃ electrocatalyst for stable and efficient acidic CO₂ conversion to synthesize formic acid (HCOOH) with low K⁺ concentration. The carbon layer protects the oxidized In species with higher intrinsic activity from reductive corrosion, and also peripherally formulates a tip-induced electric field to regulate the adverse H⁺ attraction and desirable K⁺ enrichment. In an acidic electrolyte at pH 0.94, only 0.1 M low K⁺ is required to achieve a Faradaic efficiency (FE) of 98.9% at 300 mA cm⁻² for HCOOH and a long-time stability of over100 h. By up-scaling the electrode into a 25 cm² electrolyzer setup, a total current of 7 A is recorded to sustain a durable HCOOH production of 291.6 mmol L⁻¹ h⁻¹.

The electrocatalytic carbon dioxide reduction reaction (CO₂RR) to formic acid (HCOOH) is widely recognized as a promising carbon-neutral technology for reducing carbon emissions and reserving intermittent renewable energy[1–3]. The liquid HCOOH fuel can be easily transported, serving as a low-carbon feedstock to underpin downstream chemical infrastructures with minimized carbon footprint[4]. To date, the selective conversion of CO₂ at high current densities has mainly focused on alkaline CO₂RR system, due to the maximum inhibition of competing hydrogen evolution reaction (HER) in extreme alkaline conditions (1-7 M KOH)[5–9]. However, the KOH

[1]School of Marine Science and Engineering, Hainan University, Haikou, China. [2]School of Chemistry and Chemical Engineering, State Key Laboratory of Materials Processing and Die & Mould Technology, Key Laboratory of Material Chemistry for Energy Conversion and Storage (Ministry of Education), Hubei Key Laboratory of Material Chemistry and Service Failure, Wuhan National Laboratory for Optoelectronics, Huazhong University of Science and Technology (HUST), 1037 Luoyu Rd, Wuhan, China. [3]HSE University, Moscow, Russia. [4]State Key Laboratory of Photovoltaic Science and Technology, Institute for Electric Light Sources, School of Information Science and Technology, Fudan University, Shanghai, China. [5]Hunan Joint International Research Center for Carbon Dioxide Resource Utilization, School of Physics, School of Materials Science & Engineering, Hunan Provincial Key Laboratory of Electronic Packaging and Advanced Functional Materials of Hunan Province, Central South University, Changsha, China. [6]Donostia International Physics Center (DIPC), San Sebastián-Donostia, Euskadi, Spain. [7]These authors contributed equally: Zhitong Wang, Dongyu Liu, Chenfeng Xia, Xiaodong Shi. ✉e-mail: cfxia@hust.edu.cn; byxia@hust.edu.cn; tianxl@hainanu.edu.cn

electrolyte inevitably reacts with the $CO_2$ feed to produce (bi)carbonate by-products, which lower the single-pass utilization (SPU) of $CO_2$ and degrade the hydrophobic gas-transporting channels, leading to attenuation of catalytic performance[10–13].

The development of acidic $CO_2RR$ systems is feasible to address the above issues[14,15]. The acidic electrolyte can avoid the non-reactive depletion of $CO_2$ and recycle $CO_2$ by neutralizing (bi)carbonates generated in the interfacial alkaline microenvironment[16]. Imperfectly, acidic $CO_2RR$ generally entails a tightly hydrated cation ($K^+$) layer at the outer Helmholtz plane (OHP) to repel $H^+$ and inhibit the HER competition[17]. Hence, the excessive $K^+$ can combine with (bi)carbonate anions that are untimely neutralized to form salt precipitates, causing stability concerns resembling the alkaline systems[18,19]. Some pioneering studies have explored the addition of $K^+$-free cationic organic salts to construct positively charged layers on the catalytic surface[20–22]. However, the catalytic performance is inferior to that of the $K^+$-containing $CO_2RR$ system, since the partial dehydration of $K^+$ holds the optimized adsorption capacity to polarization intermediates through short-range electrostatic interactions[23,24]. It stands to reason that a moderate $K^+$ environment is essential to harmonize the activity-stability trade-off in acidic $CO_2RR$. Another factor related to catalytic activity and stability is the anti-reduction property of the catalyst. Performant catalysts with oxidation state suffer from self-reduction under negative bias (e.g., the pristine $Cu_2O$ retained only 32.1% of its $Cu^+$ component after 20 mins of $CO_2RR$ electrolysis), inducing electronic structure redistribution of the catalytic sites and consequently compromising the catalytic efficiency[25–29]. For example, the Faradaic efficiency (FE) of $In_2O_3$ toward HCOOH was reduced to 70% within 1 h as a consequence of the unavoidable reduction of $In_2O_3$ to metallic In[30]. Taken together, the construction of a desirable acidic $CO_2RR$ system for HCOOH production depends on optimized $K^+$ environment and efficient catalyst with robust oxidation state (Fig. 1a).

Tip-induced locally enhanced electric fields have been demonstrated to enrich $K^+$ ions at the catalytic interface in alkaline system[31,32]. Such principle sheds light on employing tip-like catalysts to locally cumulate $K^+$ ions from the low $K^+$ concentration bulk phase in acidic

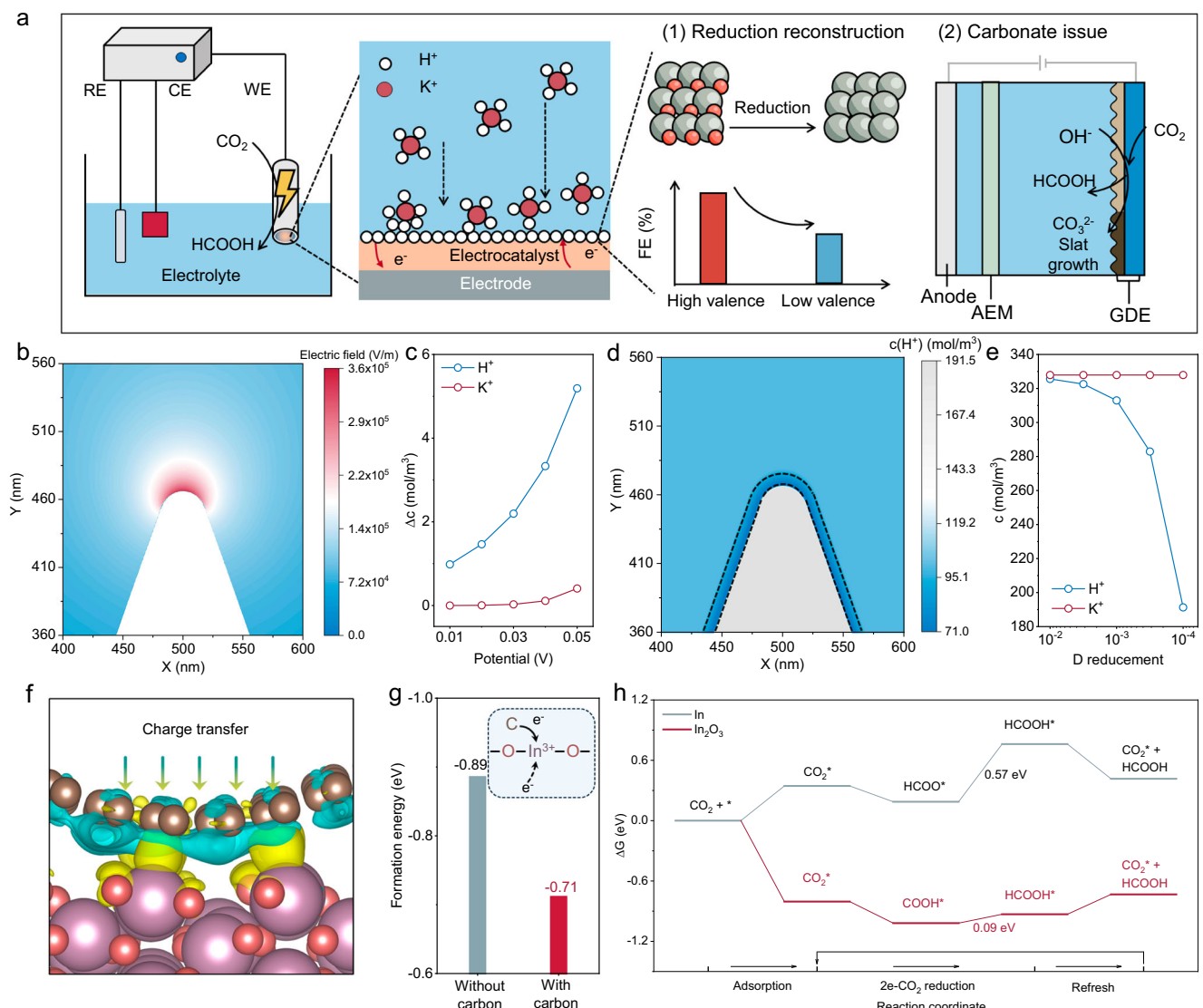

**Fig. 1 | Computational simulations screening. a** Schematic illustration of challenges in acidic $CO_2RR$. **b** Tip-induced electric field distribution. **c** Potential-dependent concentrations of $K^+$ and $H^+$ in tip-featured catalyst. **d** The effect of tip carbon coating on the distribution of $H^+$. **e** Diffusion-dependent concentrations of $K^+$ and $H^+$ in tip-featured catalyst with carbon coating. **f** Schematic modeling of the electronic interaction between the carbon layer with $In_2O_3$. Color code: purple for In, red for O, and brown for C. The blue and yellow regions indicate electron loss and gain, respectively. **g** Influence of carbon layer on oxygen vacancy generation. **h** Gibbs free energy diagrams for HCOOH on In and $In_2O_3$. The refresh step includes HCOOH desorption and $CO_2$ re-adsorption. The catalytic cycle starts from the second step as indicated by the arrows. Source data for Fig. 1b-e and Fig. 1g-h are provided as a Source Data file.

$CO_2RR$ systems, enabling efficient $CO_2$ conversion while avoiding salt precipitation. However, the enrichment effect of tip-induced electric fields is not exclusive to $K^+$[33,34]. The possibly tip-induced $H^+$ enrichment effect is generally overlooked by relevant studies, especially in acidic $CO_2RR$, which may be deviated from the original motivation of designing locally concentrated $K^+$ to restrain HER kinetics. Therefore, an external concentration field is required to cooperate with the tip-induced electric field to selectively permeate $K^+$ and throttle $H^+$ ions flux. Furthermore, the concentration field regulation module is also required to ideally protect the oxidation station of catalysts, therefore ensuring efficient and durable HCOOH synthesis in low $K^+$-mediated acidic $CO_2$ electrolysis.

In this work, theoretical calculations are performed to screen candidates with suitable concentration field modulation and oxidation state protection capabilities. Simulation modeling reveals that applying a tightly coated carbon layer on the tip-like catalyst selectively ensures tip-induced $K^+$ cumulation while eliminating the adverse $H^+$ enrichment. The robust anti-reduction property is also endowed to the encapsulated $In_2O_3$ catalyst for sustaining high HCOOH activity. To verify this proposal, the carbon coated tip-like $In_2O_3$ model catalyst is prepared by vacuum pyrolysis, which delivers a superior HCOOH FE of 98.9% at a current density of $300\,mA\,cm^{-2}$ in $H_2SO_4$ (pH = 0.94) solution with $0.1\,M\,K^+$, together with long-term catalytic stability over 100 h. Despite at a reduced current density of $50\,mA\,cm^{-2}$, the carbon coated tip-like $In_2O_3$ still manifests significant inhibition of HER with a minimized FE of only 18%, in sharp contrast to the 43% of the carbon-free counterpart. In situ characterization measurements combined with theoretical calculations demonstrate the dual-field synergy, i.e., the interfacial concentration field and the tip-induced electric field, on regulating the reaction microenvironment and catalyst oxidation state. This work provides a technically feasible and economically valuable solution to accelerate the industrialization of $CO_2RR$ technology.

## Results

### Finite element simulations and density functional theory (DFT) calculations

Here, using COMSOL Multiphysics finite element simulations, the potential of carbon layer as concentration field control modules in tip-featured catalysts was comprehensively evaluated. The tip-like structure led to an enhanced local electronic field because of the high curvature, which is expected to facilitate the adsorption of cations in the electrolyte, i.e., the $K^+$ and $H^+$ ions (Fig. 1b). Inferiorly, the concomitant attraction of $H^+$ would break the desired alkaline microenvironment benefited from $K^+$ enrichment, promoting HER and counteracting the performance improvement from the tip-like structure. Figure 1c displayed that the concentration distribution of $K^+$ and $H^+$ adjacent to the tip-like model surface were more concentrated compared to that of the slab model, exhibiting a positively proportional relation to the applied potential. As $H^+$ ions were consumed during the reaction, their migration from bulk electrolyte to catalyst surface was promoted by the electric field, making the $H^+$ accumulation more significant. Nevertheless, the electronic field was mainly confined within the electrical double layer (EDL), and the outer $H^+$ migration was determined by the concentration gradient. The carbon layer was amorphous and porous, which could block the $H^+$ migration and thus reduce its concentration near the catalytic surface (Supplementary Fig. 1). We incorporated this effect by decreasing the ion diffusion coefficients within an outer thin-shell region of the catalyst surface, where the $H^+$ concentration was effectively reduced as shown in Fig. 1d. This was reasonable since the ion diffusion coefficients in carbon materials were typically several orders of magnitude lower than those in aqueous solutions[35]. The relevant surface concentrations of $K^+$ and $H^+$ ions were plotted in Fig. 1e, which showcased the considerably mitigated $H^+$ accumulation due to carbon layer confinement, and the

concentrated $K^+$ was ideally retained. Therefore, the cooperation of tip-like structure and carbon layer confinement led to selective $K^+$ enrichment at the catalyst surface through the synergy of concentration and electric fields, which is expected to realize efficient acidic $CO_2RR$ at low $K^+$ concentrations.

The chemical interactions between the carbon layer and the $In_2O_3$ catalyst were investigated by DFT calculations (Supplementary Figs. 2–7 and Supplementary data 1). The charge density redistribution after carbon confinement was illustrated in Fig. 1f, demonstrating distinct charge transfer from carbon to $In_2O_3$. $In_2O_3$ was easily reduced to metallic In during $CO_2RR$ because of its strong electron affinity. The carbon layer served as electron doner to stabilize the $In_2O_3$ catalyst and maintain the active $In^{3+}$ reactive sites. Figure 1g indicated that the formation energy of oxygen vacancies on $In_2O_3$ was greatly increased with the presence of the carbon layer, suppressing the reduction of $In_2O_3$. The carbon atoms could also coordinate with the surface oxygen atoms to prevent the formation of oxygen vacancies. To verify the superior $CO_2RR$ catalytic activity of oxidative In sites, the corresponding reaction free energy diagram of $In_2O_3$ and In was depicted in Fig. 1h[30]. The adsorption strength of $CO_2$ on $In_2O_3$ was much stronger than that on In, favoring the reactant enrichment at the catalyst surface. Both $In_2O_3$ and In tended to produce HCOOH along two-electron $CO_2RR$, and the overall reaction was limited by the second electrochemical proton-coupled electron transfer step. The free energy change of this rate-limiting step on $In_2O_3$ was significantly lower than that on In, implying an efficient conversion of $CO_2$ to HCOOH on $In_2O_3$. Hence, the carbon layer was anticipated to maintain the high intrinsic catalytic performance of $In_2O_3$ by preventing its reduction under the $CO_2RR$ working conditions.

### Preparation and characterization of catalysts

According to theoretical insights, the desired carbon coated tip-like $In_2O_3$ was prepared by vacuum pyrolysis treatment of In-organic framework precursor (In-rho-ZMOF) (Fig. 2a). The successful preparation of In-rho-ZMOF was confirmed by the identical X-ray powder diffraction (XRD) peaks corresponding to the standard card, and its rhombic dodecahedral morphology was verified using scanning electron microscopy (SEM) (Supplementary Figs. 8–9)[36]. During the vacuum pyrolysis treatment, while maintaining the dominant rhombic dodecahedral structure, the organic ligands were unable be carbonized to form a carbon skeleton to support the internal structure due to rapid volatilization (Supplementary Fig. 10). As a result, polyhedral invagination was sharply reduced with the apical angle from ~70° to ~23°, forming carbon coated $In_2O_3$ with a distinct tip structure (denoted as Vac) (Fig. 2b, c). By switching the pyrolysis atmosphere from vacuum to Ar, In-rho-ZMOF evolved into carbon coated $In_2O_3$ with flat polyhedral structure, which fully inherited the precursor morphology without tip-like feature. To verify the role of carbon layer, the samples derived from pyrolysis of In-rho-ZMOF under vacuum and Ar atmosphere, respectively, were process by air calcination to obtain tip-like $In_2O_3$ and flat $In_2O_3$ both without the carbon layer encapsulation (denoted as Vac-air and Ar-air). The XRD patterns shown that Vac, Vac-air and Ar-air all exhibited well-defined cubic phase $In_2O_3$ structures (PDF#71-2194), indicating that the carbon layer coated on the surface of Vac-air is amorphous with no interference to the growth of $In_2O_3$ crystals (Fig. 2d). The Raman spectroscopy of Vac exhibited typical signals at $1330\,cm^{-1}$ and $1600\,cm^{-1}$, which is attributed to the presence of carbon layers (Supplementary Fig. 11). The absence of such characters for Vac-air and Ar-air indicated that the predesigned carbon layer has been completely removed by air calcination. The transmission electron microscopy (TEM) image clearly depicted the carbon coated spiky morphology of the Vac, with cavity formed inside the material causing the skeleton to shrink, accompanied by a plunge in the apical angle (Fig. 2c). Besides, amorphous carbon layers of ~10 nm thickness conformally encapsulated on $In_2O_3$ nanoparticles, which

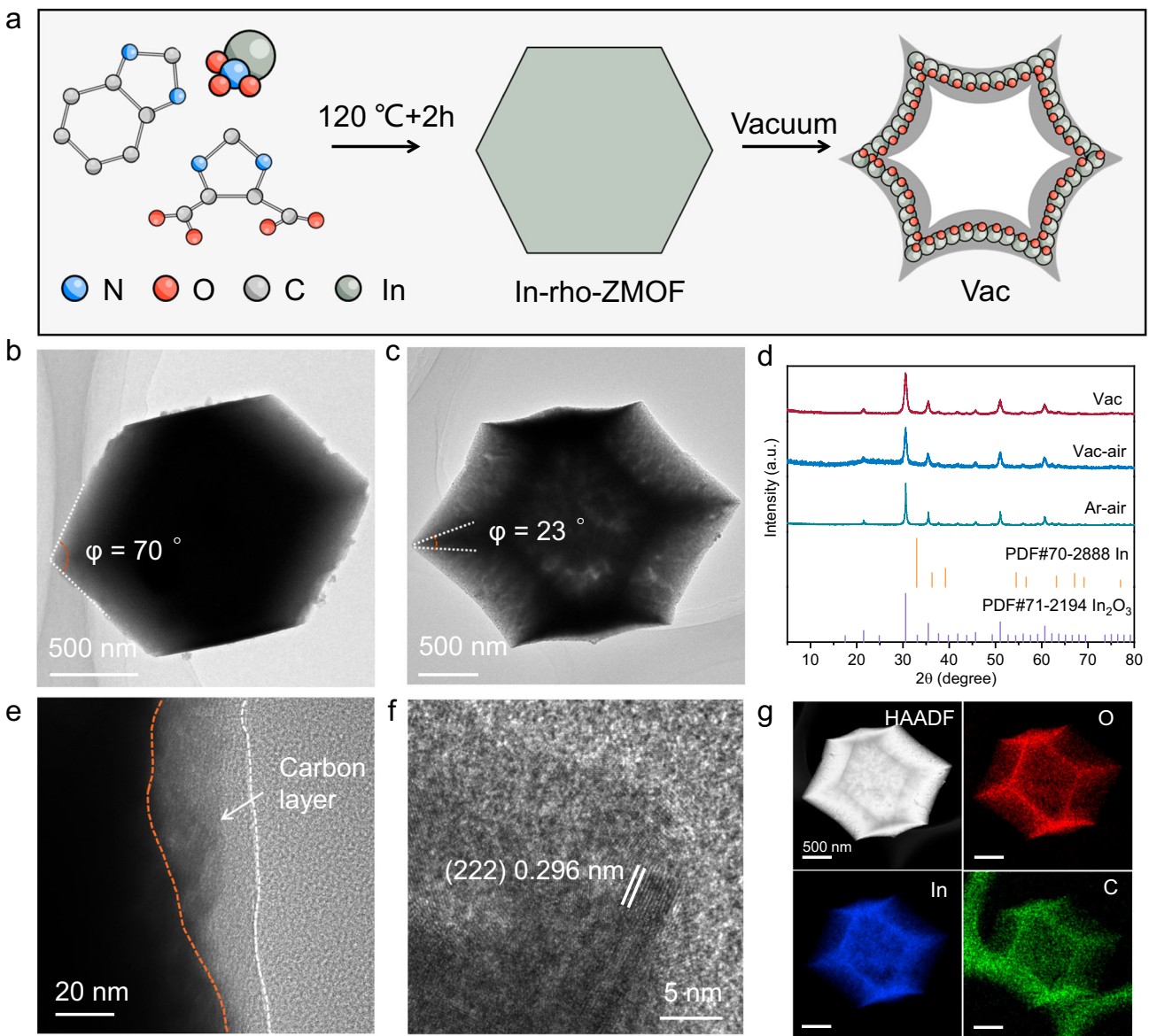

**Fig. 2 | Synthesis and structural characterization. a** Schematic preparation of Vac. TEM images of (**b**) In-rho-ZMOF and (**c**) Vac. **d** XRD patterns of Vac, Vac-air and Ar-air. **e, f** HRTEM and (**g**) EDS elemental mapping of Vac. Source data for Fig. 2 d are provided as a Source Data file.

possess a lattice spacing of 0.296 nm consistent with cubic $In_2O_3$ (222) facet (Fig. 2e, f). The morphology of Vac-air was identical to Vac, except for the disappeared carbon layer under high-resolution (HR)TEM (Supplementary Fig. 12). In contrast, cavity was not observed in Ar-air as the carbon skeleton could be effectively preserved in pyrolysis pretreatment under Ar gas (Supplementary Fig. 13). The variation in contact angle also served to corroborate the formation of the tip-like structure, wherein Vac exhibited a larger contact angle than Ar-air (Supplementary Fig. 14). This can be attributed to the smaller apical angle of the tip-like structure, which results in an increased Laplace pressure for the gas[37]. The fully inherited rhombic dodecahedral structure contrasts sharply with the distorted tip-like Vac, which may contribute to distinct catalytic performance. The elemental mapping of energy dispersive X-ray spectroscopy (EDS) intuitively visualized the edge-and-corner profile of the Vac, where the In, O and C elements were evenly distributed along the contracted framework (Fig. 2g). The above characterizations demonstrated the successful preparation of carbon coated tip-like catalysts conforming to the simulation model, and the fine tuning of morphology and composition could be implemented by switching the pyrolysis atmosphere.

## CO2RR performance

The CO2RR performance evaluation was initially performed in 0.05 M $H_2SO_4$ electrolytes (pH = 0.94) with varying $K^+$ concentrations using a flow cell to investigate the $K^+$-sensitive catalytic activity and determine the minimum $K^+$ threshold. The catalytic selectivity was evaluated by chronopotentiometry electrolysis to exclude the interference of pH fluctuations on the catalytic interface. The CO2RR gaseous and liquid products were quantitatively analyzed by gas chromatography (GC) and nuclear magnetic resonance (NMR), respectively, and detailed product distribution profiles were obtained by plotting with the applied electrolysis currents (Supplementary Figs. 15–18). In the presence of 1 M $K^+$, the HER was significantly suppressed for Vac, Vac-air and Ar-air (Fig. 3a). The maximum $H_2$ FE was less than 10.0% even at a current density of 50 mA cm$^{-2}$ and reached a minimum of 3.6% at 300 mA cm$^{-2}$. The close $H_2$ FE of different samples at the same current suggests that a concentration of 1 M $K^+$ is excessive for the acidic CO2RR system involved in this work. Furthermore, the negative correlation between $H_2$ FE and current density stems from the fact that an increased current accelerates the consumption of $H^+$, causing an elevated local pH at the reaction interface to block HER. Upon decreasing

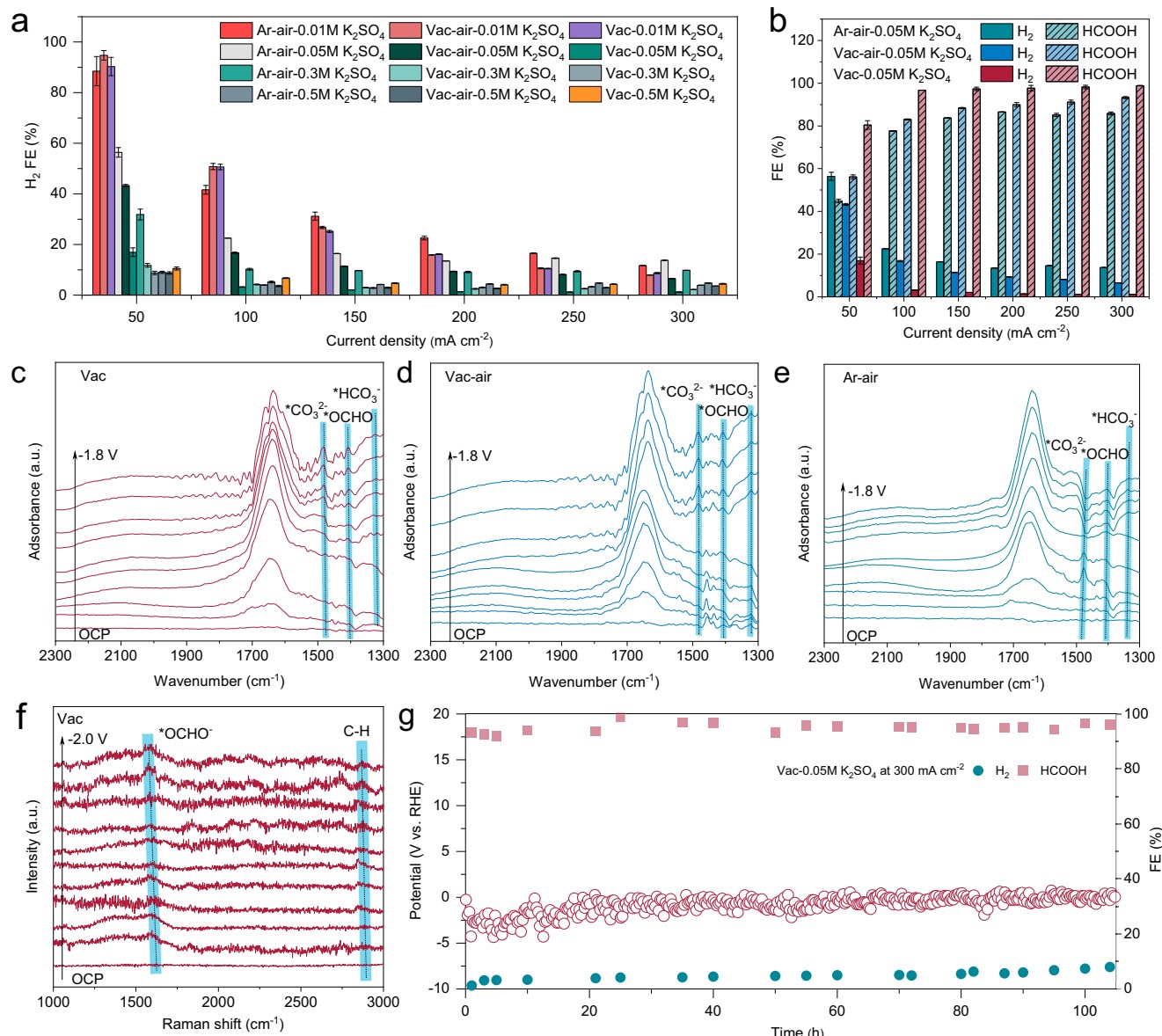

**Fig. 3 | Acidic CO₂RR performance. a** Current-dependent H₂ FE of Vac, Vac-air and Ar-air in 0.05 M H₂SO₄ with different K⁺ concentrations. **b** Current-dependent FE of Vac, Vac-air and Ar-air in 0.05 M H₂SO₄ electrolyte with 0.1 M K⁺. In situ ATR-SEIRAS spectra of (**c**) Vac, (**d**) Vac-air, (**e**) Ar-air. **f** In situ Raman spectra of Vac. **g** Catalytic stability test at 300 mA cm⁻² of Vac. Error bars represent the standard deviation of three independent measurements. The applied potentials for in situ measurements were without iR_Ω correction. Source data for Fig. 3 are provided as a Source Data file.

the K⁺ concentration to 0.6 M, the H₂ FE of the catalysts at a current density of 50 mA cm⁻² became differentiated, manifesting different K⁺ sensitivities due to structural and compositional discrepancies. It can be found that Vac shows the lowest H₂ FE of 8.8%, followed by 11.8% for Vac-air and 31.9% for Ar-air. As in the case of 1 M K⁺, the H₂ FE decreased with increasing current density, but the tendency of differential distribution among catalysts was retained even at 300 mA cm⁻². This suggested that K⁺ in H₂SO₄ was in a deficit state at this point, and further enrichment of K⁺ based on the tip effect was desired to inhibit HER. Although Vac-air also featured an analogous tip-containing structure, the HER blockage effect on Vac-air was inferior to Vac due to the difficulty of selectively shielding the tip-induced H⁺ attraction and avoiding In₂O₃ metallization. To investigate the minimum K⁺ threshold applicable to Vac-air, the K⁺ concentration was tuned down to 0.1 M. In this case, the activity of Ar-air on HER versus CO₂RR began to invert, and its H₂ FE could only be reduced from 56.3% to 13.7% as the current density increased from 50 mA cm⁻² to 300 mA cm⁻². Surprisingly, Vac

retained a strong suppressive effect on HER, with only 16.9% for H₂ and 80.5% FE for HCOOH at 50 mA cm⁻², much higher than 56.1% for Vac-air and 43.6% for Ar-air. With a slightly increased current density of 100 mA cm⁻², the H₂ FE of the Vac decayed to 3.2%, comparable to that of the Vac at 300 mA cm⁻² in 1 M K⁺ (Fig. 3b). Consequently, Vac achieved a HCOOH FE of > 96% over a wide current density range from 100 to 300 mA cm⁻² at a low K⁺ concentration of 0.1 M. The superior CO₂RR catalytic performance of Vac was also inseparable from the reasonable optimization of the carbon layer thickness. An increase or decrease in the carbon layer thickness of Vac sample (denoted as Vac-1 and Vac-2) would result in a corresponding decline in catalytic activity (Supplementary Figs. 19–20). Further reducing K⁺ to 0.02 M, the catalytic reaction of Vac catalyst with the lowest K⁺ sensitivity was dominated by HER, reaching a 90.3% FE for H₂ at 50 mA cm⁻², verifying laterally the crucial role of K⁺ for acidic CO₂RR. Therefore, 0.05 M H₂SO₄ with 0.1 M K⁺ was identified as the optimal acidic CO₂RR condition for subsequent characterization analysis.

## Insights into the catalytic mechanism

To gain in-depth information on the distribution and evolution of the reaction intermediates, in situ spectroscopy was employed for real-time monitoring the $CO_2RR$ (Supplementary Figs. 21–22). The key intermediate *OCHO was observed in the in situ attenuated total reflectance surface-enhanced infrared absorption spectroscopy (ATR-SEIRAS) of all samples at a wavelength of 1407 cm$^{-1}$ (Fig. 3c–e)[38,39]. Notably, the first appearance of *OCHO signal for Vac was recorded at −0.8 V vs. reversible hydrogen electrode (RHE), much more positive compared to −1.2 V vs. RHE for Ar-air, and the peak strength is also more intense. This suggested that $CO_2$ activation and subsequent hydrogenation was facilitated on Vac, consistent with its optimal electrocatalytic $CO_2RR$ activity. In addition, the *$CO_3^{2-}$ peak of Vac at 1482 cm$^{-1}$ intensified with the negative shift of applied potential, and the *$HCO_3^-$ at 1322 cm$^{-1}$ was hardly observed, indicating that the catalytic interface of Vac maintained locally alkaline in a strongly acidic electrolyte, which is favorable for $CO_2RR$[40]. As expected, Ar-air was incapable of reaching a steady alkaline microenvironment as the lack of local enrichment for K$^+$. The resulting *$CO_3^{2-}$ species reacted rapidly with the H$^+$ migrated from the bulk phase to form *$HCO_3^-$, as evidenced by the gradually disappeared *$CO_3^{2-}$ peaks and the intensified *$HCO_3^-$ peaks in the spectra. For Vac-air, it can be seen that the peaks of *$CO_3^{2-}$ and *$HCO_3^-$ were in a delicate equilibrium due to the tip-induced attraction of both K$^+$ and H$^+$. In situ Raman spectroscopy reaffirmed the favorable HCOOH conversion on Vac, in which the *OCHO character at 1585 cm$^{-1}$ emerged at a relatively positive potential and presented a prominent peak intensity with respect to Vac-air and Ar-air, accordant with the in situ ATR-SEIRAS results (Fig. 3f and Supplementary Fig. 23)[41]. The catalytic stability was estimated at a current density of 300 mA cm$^{-2}$. During the test, the potential of Vac exhibited slight fluctuations, and a HCOOH FE of ~99% was maintained over 100 h, demonstrating a desirable long-term stability (Fig. 3g). Upon further increasing the concentration of $H_2SO_4$ from 0.05 M to 0.1 M while maintaining the K$^+$ concentration, Vac continued to demonstrate desired catalytic performance and stability (Supplementary Fig. 24). Overall, the designed model catalysts based on computational simulations proposed a feasible approach to improve the catalytic activity and stability in low K$^+$ acidic $CO_2RR$ system.

Whether the satisfactory catalytic performance of Vac under low K$^+$ conditions is entirely due to the optimization of the local microenvironment by the carbon coated tip-featured structure. With this in mind, a series of electrochemical characterizations were conducted to reveal the deeper structure-activity relationships. The adsorption of OH$^-$ was employed as a surrogate to assess the adsorption strength of *$CO_2^{\cdot-}$ intermediate, and the more negative OH$^-$ adsorption potential of Vac inferred a stronger bonding affinity with *$CO_2^{\cdot-}$, which is also indicative of a thermodynamically favorable $CO_2$ activation process (Supplementary Fig. 25). Electrochemical impedance spectra (EIS) revealed that Vac possessed the smallest interfacial charge transfer resistance combined with a high *H coverage according to the pseudo-capacitance ($C\phi$) fitted by Nyquist plots, which meant that the activated $CO_2$ could complete the associated complex proton-coupled electron transfer steps more quickly, echoing the analyzes of in situ spectroscopy (Supplementary Figs. 26–27). Moreover, double layer capacitance measurements and catalytic activities normalized by electrochemically active specific surface area (ECSA) of the samples were performed. It could be seen that for Vac, which has the highest ECSA, still retained the optimized intrinsic activity toward HCOOH after normalization (Supplementary Figs. 28–29). Based on these considerations, it is rationalized that the desired electrocatalytic performance of Vac was boosted by simultaneous optimization of intrinsic catalytic activity and local microenvironmental.

Given the anterior theoretical calculations denotes the anti-reduction property of oxidative In sites (In$^{\delta+}$, 0 <$\delta$ <3) by carbon layer protection, the satisfactory intrinsic catalytic activity of Vac can be presumably attributed to the persistence of In$^{\delta+}$ during $CO_2RR$ (The ligand-derived bare carbon shown a negligible impact on $CO_2RR$, which further confirmed that the active site in Vac originated from In$^{\delta+}$ (Supplementary Fig. 30)). Hence, several post-reaction physicochemical measurements were performed. After the reaction, the XRD patterns of both Vac-air and Ar-air only displayed the characters of standard In (PDF#70-2888), whereas $In_2O_3$ was still observed for Vac despite the accompanying metallic In (Fig. 4a and Supplementary Fig. 31). It is suggested during $CO_2RR$, bare $In_2O_3$ suffered from severe metallization, while the carbon layer of Vac could anchor lattice O to alleviate the reductive corrosion. The In 3$d$ X-ray photoelectron spectra (XPS) results also validated the presence of In$^{\delta+}$ in Vac, whereas only In$^0$ peaks were presented for Vac-air and Ar-air (Fig. 4b and Supplementary Figs. 32–33). In addition, the O vacancy concentration of Vac was elevated after the reaction, corresponding to reductive depletion of lattice O (Fig. 4c). TEM images indicated that Vac sustained a spiky dominant structure during electrolysis, in which the majority of $In_2O_3$ nanoparticles were encapsulated by amorphous carbon layer, but accompanied with a few spherical metallic In particles ~30 nm newly emerged due to reductive reconstruction (Supplementary Fig. 34). EDS mapping found that elemental O was homogeneously distributed in the post-reaction Vac, which implies the prevented complete $In_2O_3$ metallization by carbon layer. X-ray absorption spectroscopy (XAS) measurements were employed to probe detailed valence and atomic coordination information. The In K-edge X-ray absorption near-edge structure (XANES) results of Vac showed a similar absorption edge to the $In_2O_3$ reference, inferring that the valence state of the In species is +3 (Fig. 4d). The absorption edge of Vac was negatively shifted with respect to the $In_2O_3$ reference after the reaction, but still positively positioned relative to the In reference. The valence state of the post-reaction Vac was decreased from +3 to +2.4 according to the linear combination fitting, which confirms the carbon protection to enable sufficient In$^{\delta+}$ as active sites to boost the catalytic reaction (Fig. 4e). The R space spectrum obtained by the Fourier transform (FT) of the extended X-ray absorption fine structure (EXAFS) was also analyzed. The In-O backscattering path at 1.75 Å appeared in Vac samples before and after the reaction, demonstrating the stable retention of the oxide structure (Fig. 4f). The In-O-In signal at 3.15 Å in the $In_2O_3$ reference was barely visible in Vac, which may be ascribed to the confinement of carbon layer that regulates the coordination shell of In[42]. The wavelet transform (WT) EXAFS of Vac also confirmed this result, in which the In-O-In with the intensity maxima at 9.5 Å$^{-1}$ could be probed with much lower amplitude than the In-O with the intensity maxima at 6.0 Å$^{-1}$ (Fig. 4g). In addition, a signal assigned to In-In was detected for the post-reaction Vac at 4.7 Å$^{-1}$, confirming the presence of metallic In in accordance with the XRD results. Consequently, the In K-edge EXAFS curves were fitted using the abovementioned backscattering paths to gain specific coordination information (Supplementary Fig. 35). For the Vac after the reaction, the average coordination number (CN) of In-O-In was increased from 1.2 to 3.3, and the CN of In-In was quantified to be 8.8 (Supplementary Table 1). In summary, the carbon protection of Vac successfully retained the highly active In$^{\delta+}$ sites against reductive corrosion, therefore rendering robust intrinsic catalytic ability to promote acidic $CO_2RR$.

To distinguish the different mass transfer kinetics of K$^+$ and H$^+$ within the reaction microenvironment constructed by the tip-featured structure of Vac, the linear sweep voltammetry (LSV) curves of different catalysts were recorded by rotating disk electrode (RDE) testing in 0.05 M $H_2SO_4$ + 0.05 M $K_2SO_4$. As shown in Fig. 5a, the Vac exhibited the minimum current density under Ar, illustrating the most pronounced suppression of HER. In addition, the current plateau occurred in the polarization curves under Ar in the potential range from −1.7 V to −3.0 V vs. RHE for all samples, indicating the HER is dominated by diffusion-limited H$^+$ reduction[22,43]. As the potential negatively shifted, the

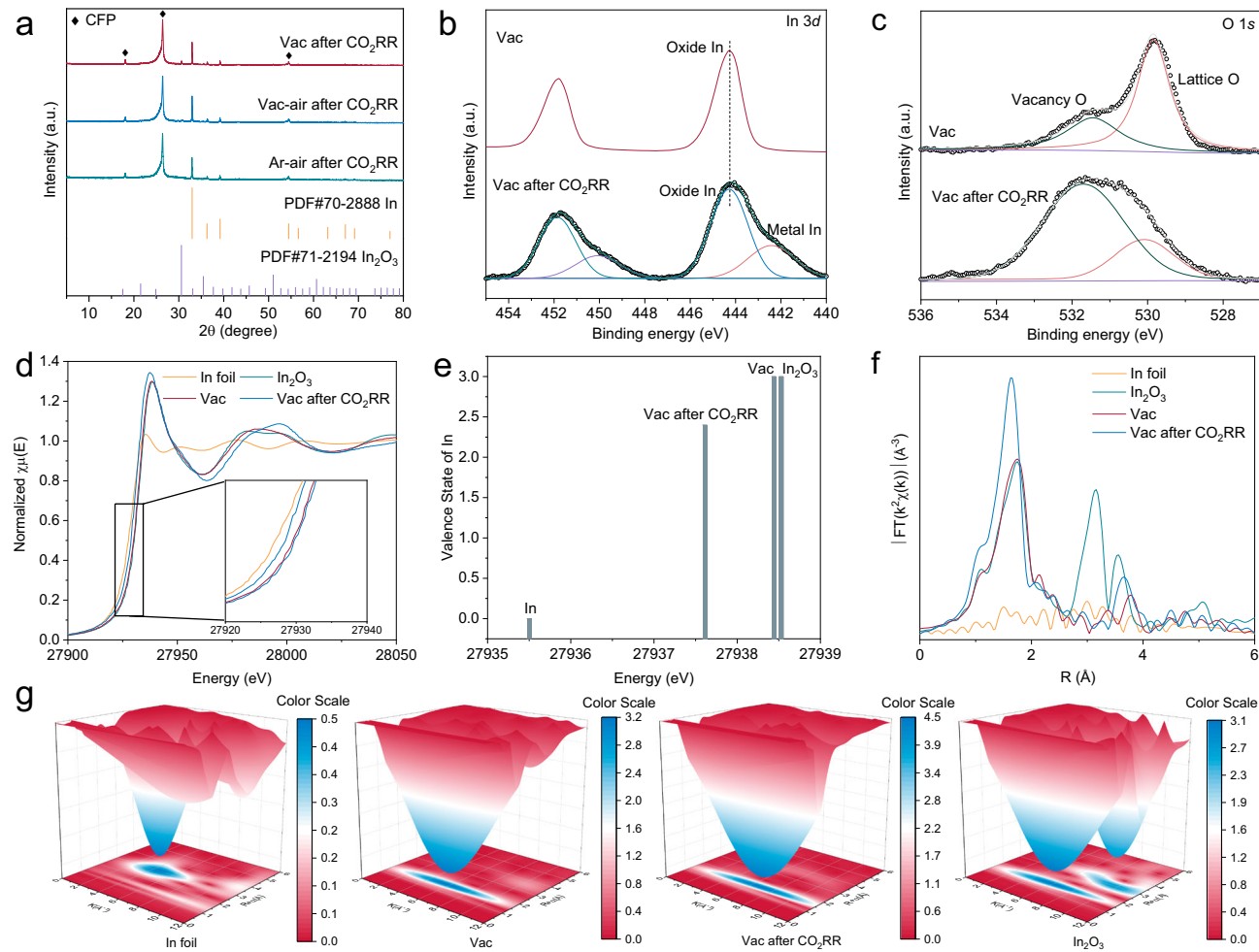

**Fig. 4 | Characterization of catalysts after CO₂RR. a** XRD patterns of Vac, Vac-air and Ar-air after CO₂RR. **b** In 3*d* XPS spectra and (**c**) O 1*s* XPS spectra of Vac before and after CO₂RR. **d** In K-edge XANES of Vac before and after CO₂RR. **e** Calculated valance state by linear combination fitting of Vac before and after CO₂RR. **f** FT-EXAFS spectra and (**g**) WT-EXAFS plots of Vac before and after CO₂RR. Source data for Fig. 4 are provided as a Source Data file.

kinetically controlled $H_2O$ reduction began to prevail and an increment in current density was achieved. To verify the underlying relation of current plateau with diffusion limitation, the rotate speed was gradually increased to accelerate the $H^+$ mass transfer. It can be seen that the current plateau of all catalysts showed an upward trend as the rotating speed increased from 900 to 2500 rpm (Supplementary Fig. 36). Therefore, a smaller current plateau represented a potent inhibition of $H^+$ diffusion, implying a reduced $H^+$ concentration and a higher pH at the catalytic interface that conducive to CO₂RR kinetics. Consequently, Vac enabled the selective enrichment of $K^+$ through the synergistic effect of the concentration field and the tip-induced electric field. The cumulation of hydrated $K^+$ in the OHP functioned as an electrostatic shield to constrain the $H^+$ transport kinetics from bulk solution to electrode surface, leading to the smallest current plateau. The maximum current plateau of Ar-air was due to the poorly enriched $K^+$ at the catalytic interface to exert HER. Particularly, for Vac-air, the tip-induced $K^+$ enrichment was partially counteracted by the simultaneous attraction of $H^+$ due to the absence of carbon layer protection, resulting in a moderate CO₂RR activity between Vac and Ar-air. The kinetic-limited current ($j_K$) and $H^+$ diffusion coefficient ($D_{H^+}$) of HER were calculated according to Koutecký–Levich and Levich equations, respectively (Fig. 5b and Supplementary Fig. 37)[20,43]. The results demonstrated that the $D_{H^+}$ of Vac was relatively 10% lower than that of Ar-air, and the corresponding $j_K$ of HER was reduced from 217 mA cm⁻² for Ar-air to 87 mA cm⁻² for Vac (Fig. 5c). Kinetic isotope effect (KIE) experiments

were conducted to feed back the $H_2O$ dissociation kinetics[40,44]. The highest KIE value of Vac (0.82) compared with Vac-air (0.59) and Ar-air (0.48) was considered for the slowest $H_2O$ dissociation originated from the ability difference to guarantee an alkaline microenvironment in acidic CO₂RR conditions (Supplementary Fig. 38).

In situ ATR-SEIRAS is recognized as an important tool for the study of interfacial $H_2O$ on catalytic surfaces. The broad peaks of the O-H stretching mode of $H_2O$ molecules in the range of 2800 cm⁻¹ to 3800 cm⁻¹ can be deconvoluted into three types of interfacial $H_2O$ species, including 4-coordinated hydrogen-bonded rigid $H_2O$ (3300 cm⁻¹), 2-coordinated hydrogen-bonded medium $H_2O$ (3480 cm⁻¹), and weakly coordinated hydrogen-bonded $K^+$ hydrated $H_2O$ ($K^+(H_2O)_n$) (3610 cm⁻¹), respectively[45,46]. Therefore, the regulatory mechanisms regarding the reaction microenvironment can be explained by investigating the real-time evolution of $K^+(H_2O)_n$ at different potentials. The $K^+(H_2O)_n$ underwent an increase in relative proportion with negatively shifted potential for both Vac and Vac-air, whereas no apparent change was observed for Ar-air, substantiating the enrichment of $K^+$ by the tip-featured structure (Fig. 5d-f). In contrast, the negatively charged electrode surface of Ar-air attracted a large amount of $H_3O^+$, which reoriented by the strong interfacial electric field to form more aligned $H_2O$ dipoles, resulting in a dramatic increase in the rigid water peak under applied bias[47]. To further understand the mechanistic difference between Vac and Vac-air in the accumulation of $K^+$, their Stark tuning behavior was investigated by taking the frequency shift of $K^+(H_2O)_n$ as a function of applied

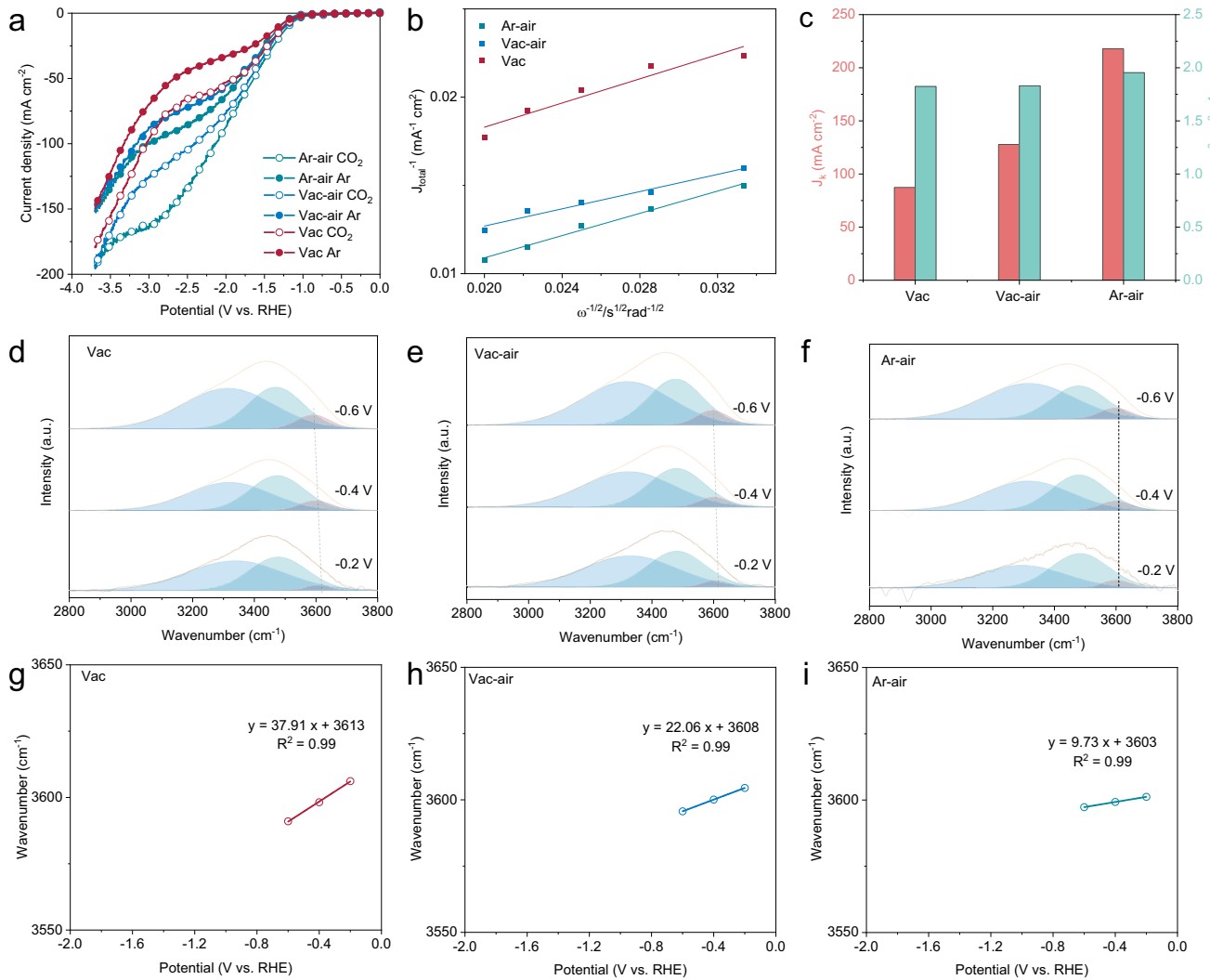

**Fig. 5 | Mechanism studies. a** LSV curves of Vac, Vac-air and Ar-air in Ar and CO$_2$-saturated electrolyte. **b** Koutecký–Levich plots of Vac, Vac-air and Ar-air at −2.8 V vs. RHE and (**c**) corresponding calculated j$_K$ and D$_{H+}$. In situ ATR-SEIRAS spectra with three O·H stretching mode through Gaussian fitting of (**d**) Vac, (**e**) Vac-air, (**f**) Ar-air. Stark tuning behavior of (**g**) Vac, (**h**) Vac-air, (**i**) Ar-air. There is no iR$_\Omega$ correction for applied potentials. Source data for Fig. 5 are provided as a Source Data file.

potential[48]. The K$^+$(H$_2$O)$_n$ vibration associated with Vac exhibited a larger Stark tuning slope of 37.9 cm$^{-1}$ V$^{-1}$ relative to Vac-air, suggesting that the K$^+$(H$_2$O)$_n$ at the Vac interface was in a stronger local potential electric field relative to Vac-air (Fig. 5g-i). Furthermore, the Stark effect of *OCHO in in situ spectroscopy was analyzed. The results indicated that as the potential became increasingly negative, the frequency shift of *OCHO in Vac also demonstrated the greatest reduction, analogous to that observed in K$^+$(H$_2$O)$_n$ (Supplementary Figs. 39–40). One possible explanation is that the electric field acting on K$^+$(H$_2$O)$_n$ for Vac-air may be partially shieled by the concurrent cumulation of H$^+$ within the EDL. As a result, the interfacial K$^+$(H$_2$O)$_n$ in Vac featured a larger dipole moment due to the selective permeation of H$^+$ through the carbon layer and demonstrated a fast response to potential switching. The above polarization curves and in situ spectroscopic analyzes confirmed the crucial role of microenvironmental regulation in the low K$^+$ acidic CO$_2$RR.

## Scale-up demonstration

Based on the great potential of Vac in low K$^+$ acidic CO$_2$RR, a scale-up demonstration was actualized to assess the large-scale electrochemical HCOOH synthesis at ampere-level processing capacity. The as-prepared Vac electrode was assembled into a homemade two-electrode electrolyzer with a reaction area of 5 * 5 cm$^2$ (Fig. 6a and Supplementary Fig. 41). Both cathode and anode chambers were fed

with 0.05 M H$_2$SO$_4$ + 0.05 M K$_2$SO$_4$ electrolyte with a flow rate of 20 mL min$^{-1}$. The CO$_2$RR performance was evaluated at applied total currents of 3 A, 5 A and 7 A (Fig. 6b). The FE of HCOOH achieved 80% when operated at 3 A with a cell voltage of 2.32 V, and further ramped up to 90% at 7 A to yield HCOOH under the cell voltage of 2.51 V (Fig. 6c). Long-term electrolysis showed that at a total applied current of 7 A, Vac could stably operate for over 15 h with merely slight fluctuations in cell voltage and HCOOH FE (Fig. 6d). The adoption of an acidic system allowed HCOOH production to be presented in the electrolyte as protonated form, bypassing the energy-intensive acidification process for product purification. The successful synthesis of HCOOH was confirmed by NMR analysis, which manifests distinguishable chemical shifts in $^1$H-NMR and $^{13}$C-NMR spectra relative to HCOO$^-$ reference (Supplementary Figs. 15 and 42). Quantitative internal standard calculations suggested that a sustained electrolysis could yield HCOOH solution with the concentration of 291.6 mmol L$^{-1}$ h$^{-1}$ (Fig. 6e)[30]. Furthermore, the SPU of CO$_2$ towards HCOOH over Vac was assessed at varying CO$_2$ flow rates. The SPU attained a maximum efficiency of 70.1% at a current density of 300 mA cm$^{-2}$ and a CO$_2$ flow rate of 3 standard cubic centimeters per minute (SCCM) (Supplementary Fig. 43). Ultimately, in terms of current density, selectivity, and stability, Vac demonstrated satisfactory performances compared to other advanced acidic CO$_2$RR catalysts, especially the feasibility to achieve

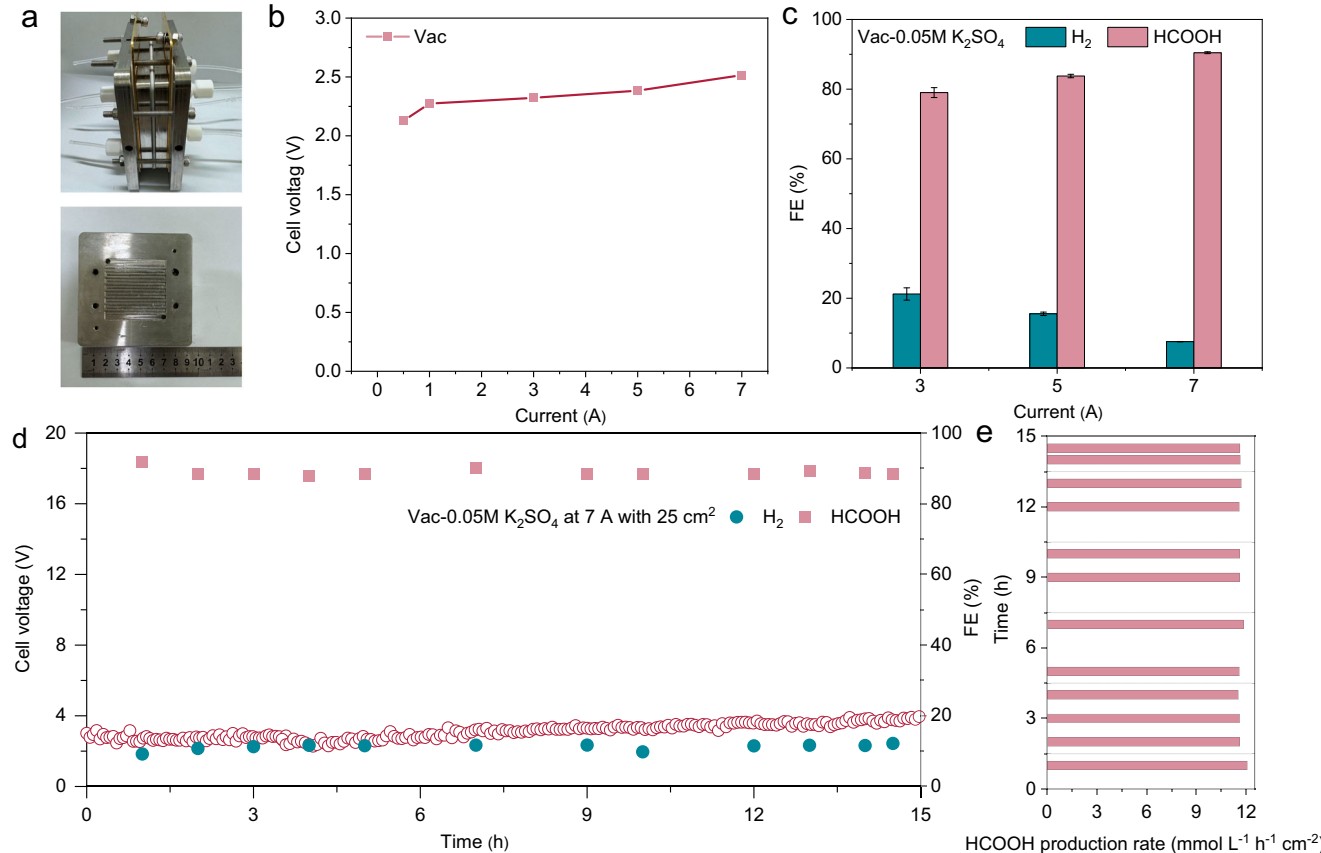

**Fig. 6 | Scale-up demonstration. a** Photograph of a scale-up electrolyzer with 25 cm². **b** Current-dependent potential plots of Vac. **c** Current-dependent FE of Vac. **d** Long-term stability test of Vac at total current of 7 A and (**e**) corresponding time-dependent HCOOH production rate. Error bars represent the standard deviation of three independent measurements. The cell voltages were compensated by $iR_\Omega$ correction with an $R_\Omega$ value of $-6.3 \pm 0.1\,\Omega$. Source data for Fig. 6b−e are provided as a Source Data file.

high CO₂RR performance at a low K⁺ concentration of only 0.1 M (Supplementary Table 2). The locally and selectively enriched K⁺ on Vac catalyst significantly reduces the operating capital cost and improves the catalytic stability, making it economically and techno-logically appealing as a practical carbon-neutral technology.

## Discussion

In summary, we demonstrated an efficient electrochemical route for synthesizing HCOOH from acidic $CO_2$ electrolysis in a low K⁺ envir-onment. Based on the cumulation effect of the tip-induced electric field for cationic species and selectively blocked H⁺ flux by carbon layer modulation, a specific enrichment of K⁺ was achieved to generate a local alkaline buffer favorable for $CO_2$ conversion in a strongly acidic (pH = 0.94) and low K⁺ (0.1 M) electrolyte. In addition, the carbon layer also strengthens the stability of oxidized In sites under reductive $CO_2$RR conditions, guaranteeing both optimized microenvironment and robust intrinsic activity to deliver superior catalytic performance. Remarkably, a high FE of 80.5% for HCOOH was delivered even at a low current density of 50 mA cm⁻², and a superior FE of 98.9% for HCOOH was maintained during operation at 300 mA cm⁻² over 100 h. The performance metrics in the demonstration of a 25 cm² scale-up reac-tion device with a total current of 7 A highlight the great promise of deploying the low K⁺ acidic $CO_2$RR toward industrialization.

## Methods
### Materials
All chemicals including indium nitrate (In(NO₃)₃, 99.9%), 4,5-imida-zoledicarboxylicacid (C₅H₄N₂O₄, 98%), N,N-dimethylformamide (DMF, 99.9%), ethanol (C₂H₅OH, 99.9%), benzimidazole (C₇H₆N₂, 99%)

were purchased from Aladdin Chemical Reagent Co., Ltd. Nafion 115 proton exchange membranes were purchased from Fuel Cell Store. High-purity argon gas (Ar, 99.999%) and carbon dioxide (CO₂, 99.99%) were purchased from Hua er Wen Gas Ltd. All materials were used without undergoing any additional purification processes.

### Preparation of In-rho-ZMOF
Indium nitrate (0.24 g), 4,5-imidazoledicarboxylicacid (0.42 g) and benzimidazole (2 g) were added sequentially into 120 mL DMF solution. Subsequently, the mixture was subjected to 10 mins of magnetic stir-ring, after which it was transferred to an oil bath maintained at 120 °C for 4 h to allow for a white precipitate formation. The precipitate was subjected to multiple washes with ethanol and DMF, respectively, and subsequently dried in a vacuum oven to yield the In-rho-ZMOF powder.

### Preparation of Vac and Vac-air
200 mg of In-rho-ZMOF was placed in a tube furnace and a vacuum pump was used to maintain a low-pressure atmosphere in the quartz tube, followed by raising the temperature (2 °C min⁻¹) to 500 °C and maintaining for 1 h. After cooling down, a pale black Vac powder was obtained. The light-yellow Vac-air powder was obtained by heating the Vac powder at 500 °C for 3 h in a muffle furnace.

### Preparation of Ar-air
200 mg of In-rho-ZMOF was placed in a tube furnace where the air inside the tube was purged with Ar gas. The In-rho-ZMOF powder was heated at 500 °C for 1 h to obtain a black powder. The above black powder was subsequently heated at 500 °C for 3 h in a muffle furnace to obtain a light-yellow Ar-air powder.

## Preparation of Vac-1, Vac-2 and bare carbon

The preparation process for Vac-1 was identical to that of Vac, with the exception being a reduction in the vacuum degree. The Vac-2 powder was obtained by heating the Vac powder at 300 °C for 1 h in a muffle furnace. The bare carbon was prepared by calcining the ligand (4,5-imidazoledicarboxylicacid and benzimidazole) in an Ar-filled tube furnace at 500 °C for 1 h.

## Characterization

X-ray diffraction (XRD) patterns were recorded using a Bruker D8 ADVANCE instrument with Cu Kα radiation. X-ray photoelectron spectroscopy (XPS) was conducted using a ThermoFisher ESCALAB 250XI instrument, which is equipped with a 200 W monochromatic Al Kα radiation and a charge neutralizer. Scanning electron microscope (SEM) was operated and collected using a Hitachi New Generation SU8010. Transmission electron microscopy (TEM) was performed using a JEOL JEM 2100 F. X-ray absorption spectroscopy (XAS) at the In K-edge of samples were performed at the XAFCA beamline of the Shanghai Synchrotron Light Source, and the data were processed and analyzed using the Demeter software package. Contact angle measurements were performed using a Dataphysics OCA20.

## Electrochemical measurements

All electrochemical measurements were carried out in ambient conditions. The acidic $CO_2RR$ was evaluated in a three-electrode flow cell configuration with Gamry Reference 3000 and Reference 30k Booster, which was constructed with a gas diffusion electrode of Sigracet 28BC (Fuel Cell Store), a KCl saturated (3.5 M) Ag/AgCl and a 1 cm × 3 cm × 1 mm Pt foil serving as the working, reference and counter electrodes, respectively. A proton exchange membrane (Nafion 115, 2 cm × 3 cm × 127 μm, Fuel Cell Store) was employed for the separation of the cathode and anode chambers after the sequentially soaked for 1 h at 80 °C in 5 wt% hydrogen peroxide, deionized water, and 5 wt% sulfuric acid, and rinsed with deionized water. The reference electrode can be selected for use by testing the open-circuit potential difference between the reference electrode and the reversible hydrogen electrode in two-electrode mode, and the potential difference can be maintained within 3 mV for 1000 s. To prepare the working electrode (1 cm × 3 cm), 10 mg of catalyst was homogeneously dispersed in a mixture containing 0.9 mL of ethanol and 100 μL of Nafion solution (5 wt%, DuPont) to form a catalyst ink, and then 300 μL of the catalyst ink was pipetted onto the Sigracet 28BC (loading: ~1 mg cm$^{-2}$) and dried overnight. The entire electrolyzer configuration was mounted and secured using polytetrafluoroethylene gaskets and controlled the reactive electrolytic area to 1 cm$^2$ (2 cm × 0.5 cm). Meanwhile, the electrochemical tests were performed on a Gamry Reference 3000 with Reference 30k Booster in a homemade two-electrode electrolyzer, and the area of the cathode gas diffusion electrode was 5 cm × 5 cm with a catalyst loading of 1 mg cm$^{-2}$, anode was the porous titanium oxide electrode with iridium oxide/ruthenium oxide, and Nafion 115 (6 cm × 6 cm × 127 μm) as the membrane. The electrolyte was circulated within the reaction chambers through a BT 100 M pump (Baoding Chuang Rui Precision Pump Co., Ltd.) at a flow rate of 10 mL min$^{-1}$. $CO_2$ was injected into the cathode gas channel using a mass flowmeter (HORIBA, S48−32) to control the flow at 20 SCCM. The electrolyte contained 0.05 M $H_2SO_4$ and various concentrations of $K_2SO_4$ at 0.01 M, 0.05 M, 0.3 M and 0.5 M, which was freshly prepared for each test. The potential was applied against the Ag/AgCl electrode and all values were transformed into reversible hydrogen electrode potential through the Nernst equation with $iR_\Omega$ compensation conversion:

$$E(vs.RHE) = E(vs.Ag/AgCl) + 0.197V + 0.0591 \times pH - 0.85 \times iR_\Omega \quad (1)$$

The electrode potentials for two-electrode electrolyzer were also $iR_\Omega$-corrected using the following formula:

$$E_{cell} = E_{applied} - 0.85 \times iR_\Omega \quad (2)$$

where the $E_{cell}$ is the compensated cell voltage, $E_{applied}$ is the actual applied cell voltage, i is the steady current density (A), $R_\Omega$ is the uncompensated resistance quantified by extrapolation of the impedance data (~4.1 ± 0.1 Ω in the three-electrode flow cell and ~6.3 ± 0.1 Ω in homemade two-electrode electrolyzer), pH is -0.94 ± 0.05 in 0.05 M $H_2SO_4$ with various $K_2SO_4$ concentrations measured with a pH meter (PHS−3E). The chronopotentiometry curves were collected in the $CO_2$ atmosphere at different current densities. The OH$^-$ adsorption measurements were conducted in an Ar-saturated 1.0 M KOH electrolyte to record the LSV curves. The ECSA was calculated through $R_f$*S. $R_f$ is the roughness factor ($C_{dl}/C_s$) and S represents the electrode active geometric area (1 cm$^2$). The value of $C_s$ is a constant of 60 μF cm$^{-2}$. $C_{dl}$ was determined by plotting the current verse the scan rate (20 to 100 mV s$^{-1}$) in CV curves, which were obtained in a 0.1 M KHCO$_3$ electrolyte with $CO_2$ saturation. EIS was recorded at varying potentials with a frequency range of 0.01 to 10$^5$ Hz and an amplitude of 10 mV to determine the resistance and pseudo-capacitance from the fitting result.

## Product analysis

The Shimadzu GC-2014 gas chromatograph was used to qualitatively and quantitatively analyze the gas phase products through the flame ionization detector (FID) and thermal conductivity detector (TCD). Faradaic efficiency (FE) of gas products in this work was calculated with the following formula:

$$FE = \frac{nFvP}{RTi} \times 100\% \quad (3)$$

In this formula, n represents the number of moles of transfer electrons for producing a gas product (CO: 2; $H_2$: 2), V denotes the measured volume concentration of gas products, v is the flow rate of $CO_2$ fed gas (mL min$^{-1}$). The value of P is 1.01 × 10$^5$ Pa, F is 96,485 C mol$^{-1}$, T is 298.15 K and R is 8.314 J mol$^{-1}$ K$^{-1}$.

The liquid product was quantitatively analyzed with a 400 MHz $^1$H-NMR and $^{13}$C-NMR spectrometer using the internal standard method. This entailed the mixing of 500 μL of electrolyte with 100 μL of deuterated water ($D_2O$), along with the addition of 50 μL of a mixed solution of dimethyl sulfoxide (5 mM) and phenol (25 mM). The FE for liquid products was determined on the formula:

$$FE = \frac{z \times F \times moles\ of\ product}{Q} \times 100\% \quad (4)$$

where the z represents the number of moles of transfer electrons for producing a liquid product (HCOOH: 2) and Q is the total electrons consumed during the whole $CO_2RR$.

The single-pass utilization (SPU) of $CO_2$ towards HCOOH was calculated as follow formula:

$$SPU = \frac{j_{total} \times FE_{HCOOH} \times 60}{2 \times F} \div \frac{v \times 1 \times P}{R \times T \times 1000} \times 100\% \quad (5)$$

where the $j_{total}$ represents the total current applied to work electrode, $FE_{HCOOH}$ is the FE for HCOOH.

## In situ ATR-SEIRAS investigation

ATR-SEIRAS spectra (Bruker Vertex70 spectrometer) equipped with a VeeMAX III ATR accessory was used for the in situ ATR-SEIRAS investigation. The counter electrode and reference electrode in the in situ

ATR-SEIRAS electrochemical cell are a Pt wire and a Ag/AgCl electrode, respectively. A gold film on the surface of a semicylindrical silicon prism substrate served as the working electrode. The electrolyte was identical to that used in the electrochemical evaluation, and a continuous $CO_2$ flow was maintained throughout the experiment period. The electrochemical cell was linked to a CHI 760E electrochemical workstation and subjected to constant potential electrolysis at varying potentials.

### In situ Raman Spectra electrochemistry investigation

A confocal Raman microscope (LabRAM HR800) equipped with a 1200 groove $mm^{-1}$ diffraction grating was used for the in situ Raman measurements. The 532 nm air-cooled laser beam was selected as the excitation source. The counter electrode and reference electrode in the in situ Raman electrochemical cell are a Pt wire and a Ag/AgCl electrode, respectively. The gas diffusion electrode sprayed with catalysts ink was used as the working electrode. The electrolyte was identical to that used in the electrochemical evaluation, and a continuous $CO_2$ flow was maintained throughout the experiment period. The electrochemical cell was linked to a CHI 760E electrochemical workstation and subjected to constant potential electrolysis at varying potentials.

### Rotating disk electrode measurement and analysis

The LSV curves were measured at varying rotating speeds in the $CO_2$ or Ar atmosphere saturated electrolyte. The kinetic current density can be obtained from the Koutecký-Levich equation:

$$\frac{1}{j_{total}} = \frac{1}{j_K} + \frac{1}{j_{plateau}} = \frac{1}{B}\omega^{-1/2} + \frac{1}{j_K} \tag{6}$$

where the $j_{total}$ represents the total current applied to work electrode, $j_K$ is the measured kinetic-limited current density of $H^+$ reduction, $j_{plateau}$ is the plateau current density of $H^+$ reduction, $\omega$ is the rotating speed of the RDE (rad $s^{-1}$), B is the proportionality coefficient.

$$B = 0.62 D_{H^+}{}^{2/3} \upsilon_{electrolyte}{}^{-1/6} nFAC^* \tag{7}$$

where $D_{H^+}$ is the diffusion coefficient of $H^+$, $\upsilon_{electrolyte}$ is the kinematic viscosity of electrolyte, n is the electron transfer number ($H_2$: 2), A is the geometric area of RDE, $C^*$ is the bulk concentration of $H^+$.

The value of $D_{H^+}$ was calculated according to Levich equation:

$$j_{plateau} = B\omega^{1/2} = 0.62 D_{H^+}{}^{2/3} \upsilon_{electrolyte}{}^{-1/6} nFAC^* \omega^{1/2} \tag{8}$$

### DFT calculation

The Vienna Ab initio Simulation Package (VASP) was employed to process all DFT calculations[49,50]. The interactions induced by electron and ion were described using the projected augmented-wave method and the Perdew–Burke–Ernzerhof functional[51,52]. The cutoff energy was established at 500 eV for the plane-wave basis set and the Gaussian smearing method with a $\sigma$ of 0.05 eV was employed. The convergence criteria of geometry optimization and charge density were 0.02 eV $Å^{-1}$ and $10^{-5}$ eV, respectively. The dispersion interaction was included using Grimme's D3 model with Becke-Johnson damping[53]. The $In_2O_3$ (110) and the In (111) surfaces were employed in our calculations[54–56]. A six-layer $In_2O_3$ (110) slab model ($1 \times 2$ supercell, 240 atoms) was used to study the interactions with the carbon layer and the formation of oxygen vacancies. The middle two layers were fixed to mimic the bulk properties. The carbon layer was represented by a monolayer graphene (120 atoms) covering the $In_2O_3$ slab. Only the gamma point was used to sample the Brillouin zone of these relatively large models. A four-layer $In_2O_3$ (110) slab model ($1 \times 1$ supercell, 80 atoms) and a four-layer In (111) slab model ($3 \times 2$ supercell, 48 atoms) were used to study

catalytic performance. The bottom two layers were fixed to mimic the bulk properties. The Brillouin zone of these relatively small models was sampled using a $2 \times 2 \times 1$ k-points mesh. Thermal corrections were carried out for adsorbed and aqueous species. All non-adsorbed species were treated as gas-phase molecules with suggested partial pressures[57]. Specifically, $CO_2$ and $H_2$ were calculated at 101325 Pa (1.0 atm). $H_2O$ was calculated at 3534 Pa according to its vapor pressure. CO was calculated at 5562 Pa according to the reported molar yield. HCOOH was calculated at a fugacity of 2.0 Pa as an ideal gas, which corresponds to an aqueous-phase activity of 0.01. This low concentration of HCOOH can properly represent most experimental conditions. However, the HCOOH desorption on $In_2O_3$ suffered from a positive free energy change, which may hinder the reaction at high HCOOH concentrations. The thermal corrections and structure visualizations were accomplished with the VASPKIT and the VESTA software packages[58].

### Finite element method

The COMSOL Multiphysics simulations were carried out with the finite element method. The 2D Nernst–Planck–Poisson equation was solved by combining the Electrostatics and the Transport of Diluted Species modules. The migration of the ions induced by the external electric field follows the Nernst–Planck equation. The accumulation of the cations near the electrode surface leads to a new electric field to compensate for this external electric field, which is described by the Poisson equation. The ion distribution and the electric potential were coupled during the simulations. Because ions are charged particles, their migration depends on the gradients of both the concentration field and the electric field. This coupling reflects the synergistic effect of the concentration and electric fields in regulating the ion distributions, which is analogous to the field synergy principle in convective heat transfer[59,60]. The electric potential difference between the bulk electrolyte and the catalyst was changed from −0.01 to −0.05 V. More negative potential not only enriche the cations near the electrode surface, but also enlarge the concentration gradient, requiring denser boundary layer meshes and making calculations more costly. We have tested the mesh independence of the current results with the electric potential difference up to −0.05 V. We fixed the potential of the catalyst, and the surface charge was calculated during the simulations. The concentrations of $H^+$, $K^+$ and $SO_4^{2-}$ at the bulk electrolyte were set to 0.1 M, corresponding to the electrolyte solution using in the catalytic system. The consumption of $H^+$ was modeled by setting a negative flux of −0.01 mol $m^{-2}$ $s^{-1}$ at the catalyst, which corresponds to a current density of 100 mA $cm^{-2}$. Considering the current density is defined with the geometric area of the electrode, we set the flux boundary condition at the bottom of the catalyst to guarantee the conceptional consistence. Moreover, we set large diffusion coefficients (10 times larger than the electrolyte diffusion coefficients) in the catalyst domain to eliminate the influences of this special treatment. The diffusion coefficients of $H^+$, $K^+$ and $SO_4^{2-}$ in the electrolyte were set to $9.31 \times 10^{-9}$, $1.96 \times 10^{-9}$, and $1.07 \times 10^{-9}$ $m^2$ $s^{-1}$, respectively. We employed an isosceles triangle with a height of 500 nm and a width of 400 nm to represent the catalyst, which is comparable to our experimental samples. The thickness of the carbon layer was set to about 10 nm. The triangle was deposed at the bottom of a 1000 nm × 1000 nm square to simulate the electrolyte. We set the ion concentrations and the electric potential of the electrolyte at the top edge of the square. The left and right sides of the square were set to insulate without flux to mimic the symmetric boundary condition.

## Data availability

The data that support the findings of this study are available within the Supplementary Information files. Source data are provided with this paper. The atomic coordinates of the optimized computational models generated in this study are provided in Supplementary Data 1. Source data are provided with this paper.

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

## Acknowledgements

This work was supported by the National Natural Science Foundation of China (22325901 (B.Y.X.), 52274297 (X.T.), 22309037 (Z.W.), 52164028 (X.T.), 22109035 (P.D.) and 22075092 (B.Y.X.)), the National Key Research and Development Program of China (2021YFA1600800 (B.Y.X.) and 2021YFA1501000 (B.Y.X.)), the Innovation and Talent Recruitment Base of New Energy Chemistry and Devices (B21003 (B.Y.X.)), Key Research and Development Project of Hainan Province (ZDYF2024SHFZO74 (P.D.)) and Start-up Research Foundation of Hainan University (KYQD(ZR)23035 (Z.W.), 20084 (P.D.), 20008 (X.T.)). Additionally, the authors acknowledge the support for comprehensive characterizations by Pico Election Microscopy Center of Hainan University and are grateful for resources from the High-Performance Computing Center of Central South University and HPC facilities at HSE University.

## Author contributions

X.T. and B.Y.X. conceived and designed the experiments. Z.W. and C.X. completed the preparation, characterization and performance evaluation of materials. D.L. and A.S.V. carried out and analyzed the DFT calculations. X.S., Y.Z., and D.Z. performed the XAS measurements and fitting. H.W., S.Z., J.L., and P.D. assisted with catalyst synthesis and data analysis. Q. L. and J. H. performed the TEM characterization and analyzed the data. Z.W., D.L., C.X., and X.S. co-wrote the paper. X.T. and B.Y.X. co-supervised the experiments. All authors discussed the results and assisted with the paper preparation.

## Competing interests

The authors declare no competing interests.
