## [Transparent Peer Review file · Nature Communications]

Tip Carbon Encapsulation Customizes Cationic Enrichment and Valence Stabilization for Low K^+ Acidic CO_2 Electroreduction

Corresponding Author: Professor Xinlong Tian

Version 0:

Reviewer comments:

Reviewer #1

(Remarks to the Author)

In their manuscript titled "Tip Carbon Encapsulation Customizes Cationic Enrichment and Valence Stabilization for Efficient Low K^+ Acidic CO_2 Electroreduction," the authors present an innovative approach to enhance the stability and electrochemical activity of the indium-based catalyst in acidic electrolytes with low K^+ concentrations, which is crucial for advancing our understanding of the CO_2 reduction reaction and for elucidating the underlying reaction mechanisms. Overall, the manuscript is well-written, and the evidence provided strongly supports the claims made. I recommend the publication in Nature Communications with the following revisions.

1. Finite Element Method: The role of the finite element method (FEM) is crucial in this study. The authors should provide a more detailed explanation of the key parameters used in the FEM analysis, including their origins and significance in the context of the work.
2. Impact of K^+ Concentration: The variation in K^+ concentration significantly influences the system's resistance, subsequently affecting the required potential for a given current. This change in applied potential can impact the selectivity of the electrocatalyst. The authors are encouraged to expand on this aspect by presenting linear sweep voltammetry (LSV) curves and electrochemical impedance spectroscopy (EIS) results across different K^+ concentrations, followed by a thorough discussion of the findings.
3. Thickness of the Carbon Layer: The authors should address whether the carbon layer's thickness influences the electrocatalyst's performance. Is there a process for optimization? Discussing this could provide valuable insights into the material's design and functionality.
4. Performance in More Acidic Environments: The stability of the electrocatalyst in 0.05 M H_2SO_4 is promising. However, the authors should investigate and report on its performance in more acidic conditions, which could be critical for practical applications.
5. Some clerical error: It should be Faradaic efficiency, not Faraday efficiency, I strongly recommend the author to check the manuscript and correct the mistakes.

Reviewer #2

(Remarks to the Author)

The manuscript proposes the design of a carbon coated tip-like In_2O_3 electrocatalyst for efficient electroreduction of CO_2 to $HCOOH$ in a strongly acidic and low K^+ electrolyte. Due to the specific enrichment of K^+ in the electrolyte and the effective retention of oxidative $In^{\delta+}$ ($0 < \delta < 3$), the prepared materials exhibited an outstanding CO_2 to $HCOOH$ selectivity of 98.9% at a current density of 300 mA cm^{-2} , as well as a long-term electrolysis capability over 100 h. The authors addressed the adverse problem of H^+ enrichment concomitant with the tip-induced K^+ concentration process by adopting the carbon layer to achieve selective penetration of K^+ , avoiding the detrimental effect of excessive H^+ ion flux on the catalytic efficiency. In situ characterization and theoretical calculations demonstrated that the carbon layer prevented the complete metallization of In_2O_3 , and the stabilized $In^{\delta+}$ sites had a superior intrinsic catalytic activity towards $HCOOH$. Overall, this study provides instructive guidance and comprehensive insights for the rational design of stable and high-performance catalysts for the acidic CO_2RR . Therefore, the reviewer suggests the acceptance of this manuscript after minor revisions according to the

comments listed below:

1. For the carbon coated In_2O_3 electrocatalyst, the possibility of the coated carbon layer acting as active sites to influence the acidic CO_2RR performance should to be considered.
2. In the HRTEM images (Fig. 2e, Supplementary Fig. S10, S11), the region marked by the dotted line and the exposed facet with corresponding lattice parameters should be labeled to make a clear comparison.
3. In the scale-up validation experiments for the 25 cm^2 electrolyzer, it is recommended that the authors exhibit more experimental details to provide context and transparency for the readers.
4. The prior work (Nat. Commun. 2024, 15, 491) suggests that the surface hydrophobicity of material is closely related to its morphology and structure. The smaller apex angle of the tip-like structure enables the increases in the Laplace pressure of the gas, thus conferring excellent hydrophobicity of the material. Therefore, the authors can further illustrate the successful construction of the tip-like structure by comparing the contact angle measurements of the counterparts.
5. The DFT calculations demonstrate that In_2O_3 possesses a much lower free energy change for electrochemical CO_2 reduction compared to In. However, the non-electrochemical HCOOH^* desorption on In_2O_3 is considerably more difficult, which may block the active sites and even deactivate the catalyst. Besides, the CO_2 adsorption on In_2O_3 is also pretty strong. Why are the species adsorbed so strongly on In_2O_3 ? Will it become a new bottleneck for CO_2 reduction?
6. In the finite element method simulations, the author claimed that "The consumption of H^+ was modeled by setting a negative flux of $-0.01 \text{ mol}/(\text{m}^2\text{s})$ at the catalyst, which corresponds to a current density of $100 \text{ mA}/\text{cm}^2$ ". However, current density is usually defined in terms of geometric area rather than the catalyst surface area which depends on the roughness. The surface area of the tip-like model can be much larger than that of the slab model. How do the authors ensure that their simulation results using different models are comparable?

Reviewer #3

(Remarks to the Author)

Tian et al. designed a new material, tip-carbon-coated In_2O_3 , for CO_2 conversion to formic acid in an acidic electrolyte. Overall, this work is innovative and well-explained, with a thoughtful combination of calculations and experiments to support the conclusion that both intrinsic activity and the local environment enhance this catalyst's performance. Before publication, I have several questions:

Based on the DFT results in Figure 1h, it appears that the desorption step is the rate-limiting step for In_2O_3 . However, the authors did not highlight the energy barrier for this step. Could the authors clarify?

In Figure 1h, the final states for In and In_2O_3 show different energies. If the reaction energy changes, can the calculations still represent the same reaction? Please explain this discrepancy.

For the DFT analysis, it would be helpful if the authors could provide all adsorption configurations to support their conclusions.

On page 12, line 259, the authors state, "Upon switching to CO_2 , its polarization curve showed the greatest increase and the current density was higher than that of Vac-air and Ar-air...". However, in Figure 5a, this doesn't seem to align with the description. The highest current density under CO_2 appears to come from the Ar-air sample, and the Vac-air sample doesn't have the highest current difference between Ar and CO_2 . Could the authors clarify this discrepancy?

On page 13, line 276, the authors mention that j_k and ΔH are plotted in Figure 5c, but they actually appear in Figure 5b. As a result, Figure 5b is not described in the text.

Reviewer #4

(Remarks to the Author)

The use of a strong acid serves as a means to enhance carbon utilization and mitigate salt precipitation in CO_2 electrolysis. However, challenges such as parasitic hydrogen evolution and electrochemical stability persist in this process. In this study, the authors introduced a new tip-like carbon-coated In_2O_3 catalyst. The carbon layer plays a crucial role in stabilizing the high-valence In^{3+} species, while an electric field induced by the tip structure helps to mitigate the adverse attraction of H^+ ions and facilitate the desired enrichment of K^+ ions. This approach was validated through theoretical analysis, a series of spectroscopic examinations, and electrochemical performance evaluations. As a result, they achieved a near-unity FE at 300 mA cm^{-2} and demonstrated an initial stability of 100 hours for formic acid production from CO_2 reduction in a strong acid environment. Furthermore, a scaled-up electrolysis employing an active electrode area of up to 25 cm^2 was successfully conducted at a total current of 7 A.

Overall, this study offers a rational catalyst design strategy to enable efficient CO_2 electrolysis in strong acid conditions. I recommend its publication in this journal following a revision. Detailed comment is provided below:

The authors stated in their abstract that "Acidic electrochemical CO_2 conversion is a promising alternative to overcome the low CO_2 utilization", yet they did not mention nor discuss this metric in the present work.

This study provides a good demonstration of enhancing CO_2RR in a strong acid environment with 0.1 M K^+ . However, the rationale behind the selection of 0.1 M K^+ concentration lacks sufficient justification. The authors assert that employing 1 M K^+ would lead to salt formation and consequently lower stability. This reasoning appears contradictory to their earlier research, where they achieved an extended stability of 5200 hours under similar conditions as reported in Nature, 2024,

626, 86-91. Please explain.

In the introduction section, the necessity of high-valence In species in promoting CO₂RR was not well explained. Instead, they simply stated that “Another factor related to catalytic activity and stability is the anti-reduction property of the catalyst. Performant catalysts with oxidation state suffer from self-reduction under negative bias, inducing electronic structure redistribution of the catalytic sites and consequently compromising the catalytic efficiency”.

The theoretical computations in this study employed two models of In and In₂O₃ to illustrate the varying catalytic performances from CO₂ to HCOOH. However, in the experimental setup, the authors prepared three catalysts, indicating a discrepancy that requires clarification.

In Figure 3c-e, the peak at ~1400 cm⁻¹ was assigned to *OCHO species. If so, the related Stark effect should be analyzed. This comment also can be applied to the Raman results in Figure 3f. Alternatively, the peak at ~1400 cm⁻¹ has been previously assigned to bicarbonate in solution (ACS Catal. 2020, 10, 8049-8057).

In Figure 4, the XRD and XAS results seem to be inconsistent. XRD showed that all samples were mostly reduced to metal-based In with a trace amount of In₂O₃ survived in Vac, whereas XAS results showed the In₂O₃-dominated phase instead of metallic In. Please explain.

Also in Figure 4, the captions of e and f were written in a wrong order.

The authors asserted that the Vac sample with carbon-coated In₂O₃ exhibited the highest performance. This raises the question of whether the real active site for CO₂RR is the In atom or the C atom. Additionally, it is important to investigate whether they considered any interface effects on CO₂RR (please refer to Nat. Energy 2020, 5, 478-486)?

The authors utilized 0.05 M H₂SO₄ as the electrolyte, lacking buffering capacity. Under high current conditions, the electrolyte's pH will notably rise. This issue is particularly critical when employing an H₂SO₄ electrolyte in a 25 cm² electrolytic cell, as maintaining a stable or acidic solution pH becomes challenging. Such pH fluctuations can significantly impact CO₂ utilization. Please explain.

Version 1:

Reviewer comments:

Reviewer #1

(Remarks to the Author)

The authors have carefully addressed my concerns, so it is now recommended.

Reviewer #2

(Remarks to the Author)

I have re-read the paper. The authors have revised their manuscript.

Reviewer #3

(Remarks to the Author)

The authors replied and revised all my questions. I recommend it to be published.

Reviewer #4

(Remarks to the Author)

The authors have taken into consideration all of my feedback. Therefore, I highly recommend its publication with just one minor suggestion. I suggest that the authors revisit the fitting baselines in Figs. 4b and 4c. Specifically, the XPS fitting baseline should be a smooth line rather than a fluctuating one.

Dear Reviewers,

Many thanks for your valuable time and work on our manuscript (**Manuscript ID: NCOMMS-24-57974A**). We have revised the manuscript carefully by supplementing additional revisions and discussions. We sincerely appreciate your comments, which have greatly improved our manuscript. Details about the revisions and our responses to the reviewers' comments are provided below in our point-by-point response.

Reviewers' comments:

Reviewer #1:

In their manuscript titled “Tip Carbon Encapsulation Customizes Cationic Enrichment and Valence Stabilization for Efficient Low K^+ Acidic CO_2 Electroreduction,” the authors present an innovative approach to enhance the stability and electrochemical activity of the indium-based catalyst in acidic electrolytes with low K^+ concentrations, which is crucial for advancing our understanding of the CO_2 reduction reaction and for elucidating the underlying reaction mechanisms. Overall, the manuscript is well-written, and the evidence provided strongly supports the claims made. I recommend the publication in Nature Communications with the following revisions.

Response: First of all, we would like to thank the reviewer for the valuable comments, and we appreciate the reviewer for recognizing the comprehensive discussion and practical significance of our manuscript. All the issues raised from the reviewer have been addressed in detail as shown in the following responses, which we hope could meet the reviewer's expectation.

Comments 1: Finite Element Method: The role of the finite element method (FEM) is crucial in this study. The authors should provide a more detailed explanation of the key parameters used in the FEM analysis, including their origins and significance in the context of the work.

Response: We appreciate the reviewer for this professional suggestion. We have supplemented a detailed

description of the methods and parameters used in our finite element method simulations in the Supporting Information. We explained the physical meaning of the governing Nernst–Planck–Poisson equations which were solved in our Multiphysics calculations and how they elucidated the mechanism of concentration and electric fields jointly regulating the ion distribution. We clarified the reasons of choosing the simulation parameters as well as the boundary conditions and provided their origins. We also discussed the significance of coupling the concentration field and the electric field in demonstrating their synergetic effects, which represents an efficient approach for modulating the properties of electrochemical reactions. We have added this part on pages S7-8 during the revision.

Revised parts in the supporting information:

Page S7-8:

The COMSOL Multiphysics simulations were carried out with the finite element method. The 2D Nernst–Planck–Poisson equation was solved by combining the Electrostatics and the Transport of Diluted Species modules. The migration of the ions induced by the external electric field follows the Nernst–Planck equation. The accumulation of the cations near the electrode surface leads to a new electric field to compensate this external electric field, which is described by the Poisson equation. The ion distribution and the electric potential were coupled during the simulations. Because ions are charged particles, their migration depends on the gradients of both the concentration field and the electric field. This coupling reflects the synergistic effect of the concentration and electric fields in regulating the ion distributions, which is analogous to the famous field synergy principle in convective heat transfer. The electric potential difference between the bulk electrolyte and the catalyst was changed from -0.01 to -0.05 V. More negative potentials not only enrich the cations near the electrode surface, but also enlarge the concentration gradient, requiring denser boundary layer meshes and making calculations more costly. We have tested the mesh independence of the current results with the electric potential difference up to -0.05 V. We fixed the potential of the catalyst and the surface charge

was calculated during the simulations. The concentrations of H^+ , K^+ and SO_4^{2-} at the bulk electrolyte were set to 100 mol/m^3 , corresponding to the electrolyte solution using in the catalytic system. The consumption of H^+ was modeled by setting a negative flux of $-0.01 \text{ mol}/(\text{m}^2\text{s})$ at the catalyst, which corresponds to a current density of 100 mA cm^{-2} . Considering the current density is defined with the geometric area of the electrode, we set the flux boundary condition at the bottom of the catalyst to guarantee the conceptual consistence. Moreover, we set large diffusion coefficients (10 times larger than the electrolyte diffusion coefficients) in the catalyst domain to eliminate the influences of this special treatment. The diffusion coefficients of H^+ , K^+ and SO_4^{2-} in the electrolyte were set to 9.31×10^{-9} , 1.96×10^{-9} , and $1.07 \times 10^{-9} \text{ m}^2/\text{s}$, respectively. We employed an isosceles triangle with the height of 500 nm and the width of 400 nm to represent the catalyst, which is comparable to our experimental samples. The thickness of the carbon layer was set to about 10 nm. The triangle was deposited at the bottom of a $1000 \text{ nm} \times 1000 \text{ nm}$ square to simulate the electrolyte. We set the ion concentrations and the electric potential of the electrolyte at the top edge of the square. The left and right sides of the square were set to insulated without flux to mimic the symmetric boundary condition.

Comments 2: Impact of K^+ Concentration: The variation in K^+ concentration significantly influences the system's resistance, subsequently affecting the required potential for a given current. This change in applied potential can impact the selectivity of the electrocatalyst. The authors are encouraged to expand on this aspect by presenting linear sweep voltammetry (LSV) curves and electrochemical impedance spectroscopy (EIS) results across different K^+ concentrations, followed by a thorough discussion of the findings.

Response: We appreciate the reviewer's insightful suggestion. It is indeed the case that alterations in the concentration of K^+ in the electrolyte can influence the solution resistance, and thus the potential necessary to achieve a given current density. This gives rise to concerns regarding the role of potential in determining the HCOOH selectivity. To address this uncertainty, we conducted a comprehensive investigation into the LSV and EIS curve responses of Vac, Vac-air and Ar-air samples at varying K^+ concentrations (0.02 M, 0.1 M, 0.6

M and 1 M). As anticipated, the solution impedance increased with the reduction of the K^+ concentration from 1 M to 0.02 M, leading to an elevation in the potential necessary to attain a given current density in the LSV (Figure R1). It is well established that the potential applied during CO_2RR can be divided into two main components: the voltage drop due to solution resistance and the working electrode potential. We were gratified to discover that the working electrode potential at a given current density for the same sample at different K^+ concentrations remained largely unchanged after IR correction. This indicates that despite the differing applied potentials for chronopotentiometry electrolysis at varying K^+ concentrations, the potential actually utilized to drive the CO_2RR reaction at the working electrode remains consistent. This alleviates concerns pertaining to product selectivity resulting from discrepancies in working electrode potential induced by alterations in solution resistance.

Figure R1. (a) LSV curves, (b) EIS and (c) solution resistance of Vac, Vac-air and Ar-air in 0.05 M H_2SO_4 with different K^+ concentrations.

Comments 3: Thickness of the Carbon Layer: The authors should address whether the carbon layer's thickness influences the electrocatalyst's performance. Is there a process for optimization? Discussing this could provide valuable insights into the material's design and functionality.

Response: We are grateful to the reviewer for this constructive and professional comment. The carbon-rich Vac-1 and carbon-less Vac-2 were prepared by modifying the pyrolysis conditions (Figure S19). The acidic CO_2RR performance was evaluated in 0.05 M H_2SO_4 electrolyte with 0.1 M K^+ , with the aim of elucidating

the mechanism by which the carbon layer thickness affects the catalytic performance (**Figure S20**). The results demonstrated that the catalytic performance of Vac-2 was superior to that of Vac-air and was surpassed by Vac. Similarly, the catalytic performance of Vac-1 was more effective than that of Ar-air and was outperformed by Vac-air. The Vac-2 sample, despite exhibiting a comparable tip-like structure to Vac, displayed a diminished catalytic performance relative to Vac. This was attributed to the absence of a sufficient carbon layer, which is essential for maintaining the stability of the dominant In_2O_3 active species in dynamic reactions and for selective regulation of cation enrichment at the tip-like structure. However, in comparison to Vac-air, which was fully metallized throughout the reaction and was unable to circumvent the unfavorable enrichment of H^+ caused by the tip-induced electric field, the enhanced catalytic activity of Vac-2 was reasonable. Vac-1 sample featured a thicker carbon layer coating, though this was achieved at the expense of tip-like structure formation. It should be noted that in the vacuum pyrolysis synthesis strategy employed in this work, the carbon layer coating and the tip-like structure construction were conducted concurrently. Consequently, it is challenging to augment the thickness of the carbon layer without compromising the integrity of the tip-like structure. Specifically, an increase in the thickness of the carbon layer necessitates a reduction in the vacuum degree. In the event of an insufficient vacuum environment, the precursor material would be unable to fully collapse and form a tip-like structure with a small apical angle. (The segmented preparation process allowed for the preparation of less carbon Vac-2 without compromising the tip-like structure. Both Vac-2 and Vac-air samples were derived from Vac sample, and the carbon layer thickness was adjusted by post-treatment with air calcination. In the preparation of Vac-2, a lower air calcination temperature and time were employed than in the Vac-air prepare process, resulting in a reduction in the carbon layer thickness without affecting the tip structure.) Therefore, although Vac-1 with a thicker carbon layer could prevent the complete reduction of In_2O_3 species, the lack of a tip-like structure rendered the carbon layer ineffective in functionalizing the microenvironment concentration field, ultimately contributing to the catalytic activity located between Vac-air and Ar-air.

In conclusion, these results demonstrated that the thickness of the carbon layer was a critical factor influencing the catalytic performance of acidic CO₂RR. Insufficient coverage of the carbon layer was unable to effectively stabilize the active oxidized In sites and regulate the selective enrichment of K⁺. However, an excessively thick carbon layer impeded the formation of the tip-like structure and was detrimental to the construction of a local alkaline environment, which was conducive to CO₂RR in an acidic environment. The results of the control group experiment between Vac, Vac-air and Ar-air in the original manuscript also supported the crucial role of the carbon layer. To avoid unnecessary repetition and ensure logical consistency, a detailed optimization explanation of the carbon layer thickness has not been added in the revised manuscript.

Revised parts in the manuscript:

Page 9:

The superior CO₂RR catalytic performance of Vac was also inseparable from the reasonable optimization of the carbon layer thickness. An increase or decrease in the carbon layer thickness of Vac sample would result in a corresponding decline in catalytic activity (Figure S19-20).

Revised parts in the supporting information:

Page S2-3:

Preparation of *Vac-1*: The preparation process for Vac-1 was identical to that of Vac, with the exception being a reduction in the vacuum degree.

Preparation of *Vac-2*: The Vac-2 powder was obtained by heating the Vac powder at 300 °C for 1 h in a muffle furnace.

Page S27-28:

Figure S19. TEM images of (a) Vac-1 and (c) Vac-2. HRTEM images of (b) Vac-1 and (d) Vac-2.

Figure S20. Current-dependent potential plots of (a) Vac-1 and (c) Vac-2. Current-dependent FE of (b) Vac-1 and (d) Vac-2.

Comments 4: Performance in More Acidic Environments: The stability of the electrocatalyst in 0.05 M H₂SO₄ is promising. However, the authors should investigate and report on its performance in more acidic conditions, which could be critical for practical applications.

Response: We are grateful to the reviewer for this insightful comment. In order to verify the CO₂RR performance of the Vac sample in a super acid electrolyte closer to practical application, we further increased the H₂SO₄ concentration from 0.05 M to 0.1 M and kept the K⁺ concentration unchanged at 0.1 M. It is encouraging to note that Vac displayed excellent acidic CO₂RR catalytic activity, comparable to that observed in 0.05 M H₂SO₄, even under the harsh conditions of 0.1 M H₂SO₄. The results of the stability test demonstrated that during a continuous electrolysis period of 30 h, the Vac sample was capable of converting CO₂ to HCOOH at an electric current density of 200 mA cm⁻² and an excellent selectivity of over 96% under electrolysis conditions of 0.1 M H₂SO₄ + 0.1 M K⁺ with negligible voltage fluctuations (**Figure S24**). The above conclusions provide compelling evidence of the significant potential of Vac samples for large-scale application and promotion. A performance description of Vac under more acidic conditions has been incorporated into the revised manuscript.

Revised parts in the manuscript:

Page 10:

Upon further increasing the concentration of H₂SO₄ from 0.05 M to 0.1 M while maintaining the K⁺ concentration, Vac continued to demonstrate excellent catalytic performance and stability (Figure S24).

Revised parts in the supporting information:

Page S32:

Figure S24. (a) Current-dependent potential plots, (b) current-dependent FE and long-term stability test of Vac in 0.1 M H₂SO₄ with 0.1 M K⁺.

Comments 5: Some clerical error: It should be Faradaic efficiency, not Faraday efficiency, I strongly recommend the author to check the manuscript and correct the mistakes.

Response: We are grateful to the reviewer for this helpful comment and apologize for this clerical error. We have thoroughly checked the manuscript and rectified these errors.

Reviewer #2:

The manuscript proposes the design of a carbon coated tip-like In_2O_3 electrocatalyst for efficient electroreduction of CO_2 to HCOOH in a strongly acidic and low K^+ electrolyte. Due to the specific enrichment of K^+ in the electrolyte and the effective retention of oxidative $\text{In}^{\delta+}$ ($0 < \delta < 3$), the prepared materials exhibited an outstanding CO_2 to HCOOH selectivity of 98.9% at a current density of 300 mA cm^{-2} , as well as a long-term electrolysis capability over 100 h. The authors addressed the adverse problem of H^+ enrichment concomitant with the tip-induced K^+ concentration process by adopting the carbon layer to achieve selective penetration of K^+ , avoiding the detrimental effect of excessive H^+ ion flux on the catalytic efficiency. In situ characterization and theoretical calculations demonstrated that the carbon layer prevented the complete metallization of In_2O_3 , and the stabilized $\text{In}^{\delta+}$ sites had a superior intrinsic catalytic activity towards HCOOH . Overall, this study provides instructive guidance and comprehensive insights for the rational design of stable and high-performance catalysts for the acidic CO_2RR . Therefore, the reviewer suggests the acceptance of this manuscript after minor revisions according to the comments listed below:

Response: We sincerely thank the reviewer for the very positive feedback of our work, and we are also grateful for the reviewer's recommendation to publish our manuscript. We have carefully improved our manuscript according to these valuable suggestions, the detailed responses are provided as below. We hope our efforts have clarified all the concerns and meet the reviewer's expectations.

Comments 1: For the carbon coated In_2O_3 electrocatalyst, the possibility of the coated carbon layer acting as active sites to influence the acidic CO_2RR performance should to be considered.

Response: We appreciate the reviewer for this professional comment. The carbon-based metal-free electrocatalysts (C-MFECs) are promising options with cost-effectiveness for large-scale electrocatalytic applications. However, for C-MFECs employed in CO_2RR , the products are mainly H_2 , as verified by many studies (*Adv. Energy Mater.* **2017**, 7, 1700759; *Angew. Chem. Int. Ed.* **2018**, 57, 13135; *Small Methods* **2021**,

5, 2001039). The electrocatalytic CO₂RR performance of bare carbon is tested for comparison, which is derived from calcining the ligand (4,5-imidazoledicarboxylic acid and benzimidazole) at 500°C for 1 h under Ar atmosphere. The resulting products are mainly H₂ (**Figure S30**). Therefore, the catalytic performance contribution of carbon in carbon coated In₂O₃ electrocatalyst could be eliminated.

Revised parts in the manuscript:

Page 11:

The ligand-derived bare carbon shown a negligible impact on CO₂RR, which further confirmed that the active site in Vac originated from In^{δ+} (Figure S30).

Revised parts in the supporting information:

Page S3:

Preparation of *bare carbon*: The bare carbon was prepared by calcining the ligand (4,5-imidazoledicarboxylic acid and benzimidazole) in an Ar-filled tube furnace at 500 °C for 1 h.

Page S38:

Figure S30. (a) LSV curves and (b) current-dependent H₂ FE of bare carbon in 0.05 M H₂SO₄ with 0.1 M K⁺ under CO₂ atmosphere.

Comments 2: In the HRTEM images (Fig. 2e, Supplementary Fig. S10, S11), the region marked by the dotted line and the exposed facet with corresponding lattice parameters should be labeled to make a clear comparison.

Response: We appreciate the reviewer for the professional viewpoint. The above figures have been revised to include the corresponding region labels and lattice parameter.

Revised parts in the manuscript:

Page 20:

Figure 2. (a) Schematic preparation of Vac. TEM images of (b) In-rho-ZMOF and (c) Vac. (d) XRD patterns of Vac, Vac-air and Ar-air. (e-f) HRTEM and (g) EDS elemental mapping of Vac.

Revised parts in the supporting information:

R12/R44

Figure S12. (a, **b**) TEM images and (c) EDS elemental mapping of Vac-air.

Figure S13. (a, **b**) TEM images and (c) EDS elemental mapping of Ar-air.

Comments 3: In the scale-up validation experiments for the 25 cm² electrolyzer, it is recommended that the authors exhibit more experimental details to provide context and transparency for the readers.

Response: We are grateful to the reviewer for this helpful comment. In the revised Supporting Information, we have added details of the construction of the two-electrode large-scale electrolyzer device and the operating configuration, in the hope of facilitating a deeper understanding of the experimental details among the readerships (**Figure S41**).

Revised parts in the supporting information:

Page S49:

Figure S41. Photographs of the two-electrode scale-up electrolyzer configuration.

R14/R44

Comments 4: The prior work (Nat. Commun. 2024, 15, 491) suggests that the surface hydrophobicity of material is closely related to its morphology and structure. The smaller apex angle of the tip-like structure enables the increases in the Laplace pressure of the gas, thus conferring excellent hydrophobicity of the material. Therefore, the authors can further illustrate the successful construction of the tip-like structure by comparing the contact angle measurements of the counterparts.

Response: We are grateful to the reviewer for their professional assessment. The contact angle of three samples was measured, and a clear trend in the contact angle distribution was identified, which was found to be correlative with the tip-like structure (**Figure S14**). Vac and Vac-air with tip-like structures exhibited similar contact angle values and were greater than those of Ar-air without tip-like structures. This is consistent with previous work reporting that the smaller apex angle of the tip-like structure increases the Laplace pressure of the gas, thereby rendering the material with superior hydrophobicity. We have supplemented the statements in the revised manuscript regarding the successful construction of the pointed structure as confirmed by the contact angle.

Revised parts in the manuscript:

Page 7:

The variation in contact angle also served to corroborate the formation of the tip-like structure, wherein Vac exhibited a larger contact angle than Ar-air (Figure S14). This can be attributed to the smaller apical angle of the tip-like structure, which results in an increased Laplace pressure for the gas.

Revised parts in the supporting information:

Page S3:

Contact angle measurements were performed using a Dataphysics OCA20.

Page S22:

Figure S14. Contact angle measurements of (a) Vac, (b) Vac-air and (c) Ar-air.

Comments 5: The DFT calculations demonstrate that In_2O_3 possesses a much lower free energy change for electrochemical CO_2 reduction compared to In. However, the non-electrochemical HCOOH^* desorption on In_2O_3 is considerably more difficult, which may block the active sites and even deactivate the catalyst. Besides, the CO_2 adsorption on In_2O_3 is also pretty strong. Why are the species absorbed so strongly on In_2O_3 ? Will it become a new bottleneck for CO_2 reduction?

Response: We appreciate the reviewer for the professional viewpoint. The strong CO_2 and HCOOH adsorptions can be attributed to the In-O coordination interactions which satisfy the octahedral configuration of unsaturated In^{3+} ions on the In_2O_3 (110) surface. These observations are consistent with the previously reported results (*J. Phys. Chem. C* **2012**, *116*, 7817–7825). Metallic In can not provide such hybridized orbitals to coordinate with oxygen ions, leading to relatively weak interactions. The HCOOH^* desorption was considered as the following reaction

The free energy change of this procedure depends on the chemical potential of HCOOH molecule in the solution. We calculated the chemical potential of aqueous HCOOH at an aqueous-phase activity of 0.01, which is a relatively standard condition as suggested (*Energy Environ. Sci.* **2010**, *3*, 1311-1315). In our experiments, fresh electrolyte was continually injected into the reaction cell. This operation ensured the concentration of HCOOH in the electrolytic cell was always close to zero, making the chemical potential of aqueous HCOOH considerably low according to the Nernst equation. Under this condition, the HCOOH desorption is expected

to be thermodynamically favorable. However, if HCOOH is accumulated in the electrolyte, the overall reaction would be limited by its desorption. We have clarified this issue and provided the method of calculating chemical potentials of non-adsorbed species on page S7 of the Supporting Information during the revision.

Revised parts in the supporting information:

Page S7:

Thermal corrections were carried out for adsorbed and aqueous species. All non-adsorbed species were treated as gas-phase molecules with suggested partial pressures. Specifically, CO₂ and H₂ were calculated at 101325 Pa (1 atm). H₂O was calculated at 3534 Pa according to its vapor pressure. CO was calculated at 5562 Pa according to the reported molar yield. HCOOH was calculated at a fugacity of 2.0 Pa as an ideal gas, which corresponds to an aqueous-phase activity of 0.01. This low concentration of HCOOH can properly represent most experimental conditions. However, the HCOOH desorption on In₂O₃ suffered from a positive free energy change, which may hinder the reaction at high HCOOH concentrations.

Comments 6: In the finite element method simulations, the author claimed that “The consumption of H⁺ was modeled by setting a negative flux of -0.01 mol/(m²s) at the catalyst, which corresponds to a current density of 100 mA/cm²”. However, current density is usually defined in terms of geometric area rather than the catalyst surface area which depends on the roughness. The surface area of the tip-like model can be much larger than that of the slab model. How do the authors ensure that their simulation results using different models are comparable?

Response: We appreciate the reviewer for this insightful comment. The tip-like catalyst was represented by an isosceles triangle in our 2D simulations. To guarantee the conceptual consistence, we set the negative mass flux of H⁺ at the bottom of the triangle catalyst region. We fixed the electric potential of the catalyst. The H⁺ ions were attracted to catalyst surface by the electrostatic interactions and crossed it due to the mass transfer. Nevertheless, the mass transfer within the catalyst is unreal since they should be consumed at the catalyst

surface already. To eliminate this effect, we set large diffusion coefficients within the catalyst region to reduce the concentration differences. This approach ensures the integral of the H^+ flux on the catalyst surface is identical to the integral of the H^+ flux calculated from the current density. We have clarified this point on pages S7-8 of the Supporting Information during the revision.

Revised parts in the supporting information:

Page S7-8:

The COMSOL Multiphysics simulations were carried out with the finite element method. The 2D Nernst–Planck–Poisson equation was solved by combining the Electrostatics and the Transport of Diluted Species modules. The migration of the ions induced by the external electric field follows the Nernst–Planck equation. The accumulation of the cations near the electrode surface leads to a new electric field to compensate this external electric field, which is described by the Poisson equation. The ion distribution and the electric potential were coupled during the simulations. Because ions are charged particles, their migration depends on the gradients of both the concentration field and the electric field. This coupling reflects the synergistic effect of the concentration and electric fields in regulating the ion distributions, which is analogous to the famous field synergy principle in convective heat transfer. The electric potential difference between the bulk electrolyte and the catalyst was changed from -0.01 to -0.05 V. More negative potentials not only enrich the cations near the electrode surface, but also enlarge the concentration gradient, requiring denser boundary layer meshes and making calculations more costly. We have tested the mesh independence of the current results with the electric potential difference up to -0.05 V. We fixed the potential of the catalyst and the surface charge was calculated during the simulations. The concentrations of H^+ , K^+ and SO_4^{2-} at the bulk electrolyte were set to 100 mol/m^3 , corresponding to the electrolyte solution using in the catalytic system. The consumption of H^+ was modeled by setting a negative flux of $-0.01 \text{ mol}/(\text{m}^2\text{s})$ at the catalyst, which corresponds to a current density of 100 mA cm^{-2} . Considering the current density is defined with the geometric area of the electrode,

we set the flux boundary condition at the bottom of the catalyst to guarantee the conceptual consistence. Moreover, we set large diffusion coefficients (10 times larger than the electrolyte diffusion coefficients) in the catalyst domain to eliminate the influences of this special treatment. The diffusion coefficients of H^+ , K^+ and SO_4^{2-} in the electrolyte were set to 9.31×10^{-9} , 1.96×10^{-9} , and $1.07 \times 10^{-9} \text{ m}^2/\text{s}$, respectively. We employed an isosceles triangle with the height of 500 nm and the width of 400 nm to represent the catalyst, which is comparable to our experimental samples. The thickness of the carbon layer was set to about 10 nm. The triangle was deposited at the bottom of a $1000 \text{ nm} \times 1000 \text{ nm}$ square to simulate the electrolyte. We set the ion concentrations and the electric potential of the electrolyte at the top edge of the square. The left and right sides of the square were set to insulated without flux to mimic the symmetric boundary condition.

Reviewer #3:

Tian et al. designed a new material, tip-carbon-coated In₂O₃, for CO₂ conversion to formic acid in an acidic electrolyte. Overall, this work is innovative and well-explained, with a thoughtful combination of calculations and experiments to support the conclusion that both intrinsic activity and the local environment enhance this catalyst's performance. Before publication, I have several questions:

Response: We sincerely thank the reviewer for the valuable time and high evaluation of our work. After revision according to your professional suggestions, we believe that the new insights will strengthen the revised manuscript, and our point-by-point responses are as follows.

Comments 1: Based on the DFT results in Figure 1h, it appears that the desorption step is the rate-limiting step for In₂O₃. However, the authors did not highlight the energy barrier for this step. Could the authors clarify?

Response: Thank you for pointing out this important issue. The free energy change of HCOOH desorption in Figure 1h was calculated as the following reaction:

where the adsorbed HCOOH* was replaced by a CO₂ molecule and became free in the solution. Actually, in the response to your second comment, we have changed the name of this step from “desorption” to “refresh”, which is a more reasonable description of this procedure. The free energy change here depends on the concentration of aqueous HCOOH, because it determines the chemical potential of HCOOH according to the Nernst equation. The aqueous HCOOH was calculated with a low aqueous-phase activity of 0.01, which is a relatively standard condition for such dilute solutions (*Energy Environ. Sci.* **2010**,*3*, 1311-1315). However, in our experiments, fresh electrolyte was continually injected into the reaction cell. This operation ensured the concentration of HCOOH in the electrolytic cell was always close to zero, making the chemical potential of aqueous HCOOH considerably lower than our current result. Under this condition, the HCOOH desorption is

expected to be thermodynamically favorable. Nevertheless, as you pointed out, HCOOH desorption would become the rate-limiting step for In_2O_3 if HCOOH is accumulated in the electrolyte, which needs to be addressed by updating the design of the reaction system. The specific value of the energy barrier depends on the concentration of aqueous HCOOH which can hardly be identified, but this positive free energy change clearly demonstrates this problem. We have clarified that the desorption step can hinder the reaction at high HCOOH concentrations and supplemented the methods of calculating the chemical potentials of aqueous species in the Supporting Information during the revision.

Revised parts in the supporting information:

Page S7:

Thermal corrections were carried out for adsorbed and aqueous species. All non-adsorbed species were treated as gas-phase molecules with suggested partial pressures.^[15] Specifically, CO_2 and H_2 were calculated at 101325 Pa (1 atm). H_2O was calculated at 3534 Pa according to its vapor pressure. CO was calculated at 5562 Pa according to the reported molar yield. HCOOH was calculated at a fugacity of 2.0 Pa as an ideal gas, which corresponds to an aqueous-phase activity of 0.01. This low concentration of HCOOH can properly represent most experimental conditions. However, the HCOOH desorption on In_2O_3 suffered from a positive free energy change, which may hinder the reaction at high HCOOH concentrations.

Comments 2: In Figure 1h, the final states for In and In_2O_3 show different energies. If the reaction energy changes, can the calculations still represent the same reaction? Please explain this discrepancy.

Response: We are sorry for the misleading expression in our previous manuscript. The catalytic cycle actually started from the second reaction step after CO_2 adsorption. We included CO_2 adsorption in the free energy diagram to show the interaction between CO_2 and In_2O_3 is much stronger than the interaction between CO_2 and In, indicating the CO_2 reactant can be enriched near the In_2O_3 surface. This significant difference also demonstrates that including the CO_2 adsorption step is highly important for calculating the free energy changes

of the subsequent reactions. At the last reaction step, the adsorbed HCOOH^* was replaced by a non-adsorbed CO_2 , and the catalytic cycle restarted from the adsorbed CO_2^* , i.e., the second reaction step. We have changed the name of the last reaction step from “desorption” to “refresh” in **Figure 1h** and explained this in the caption “The refresh step includes HCOOH desorption and CO_2 re-adsorption. The catalytic cycle starts from the second step as indicated by the arrows.” during the revision. The free energy changes from CO_2^* to the HCOOH , i.e., from the second intermediate to the final state, are the same on both surfaces.

Revised parts in the manuscript:

Page 19:

Figure 1. (a) Schematic illustration of challenges in acidic CO_2RR . (b) Tip-induced electric field distribution. (c) Potential-dependent concentrations of K^+ and H^+ in tip-feathered catalyst. (d) The effect of tip carbon

coating on the distribution of H^+ . (e) Diffusion-dependent concentrations of K^+ and H^+ in tip-featured catalyst with carbon coating. (f) Schematic modelling of the electronic interaction between the carbon layer with In_2O_3 . (g) Influence of carbon layer on oxygen vacancy generation. (h) Gibbs free energy diagrams for HCOOH on In and In_2O_3 . The refresh step includes HCOOH desorption and CO_2 re-adsorption. The catalytic cycle starts from the second step as indicated by the arrows.

Comments 3: For the DFT analysis, it would be helpful if the authors could provide all adsorption configurations to support their conclusions.

Response: Thank you for this valuable suggestion. We have provided the adsorption configurations of all reaction intermediates on In (111) surface and In_2O_3 (110) surface in the Supporting Information as **Figures S4 and S5** during the revision.

Revised parts in the supporting information:

Page S12-13:

Figure S4. Adsorption configurations of reaction intermediates on In (111) surface. The color code is the same as Figure S2.

Figure S5. Adsorption configurations of reaction intermediates on In_2O_3 (110) surface. The color code is the same as Figure S2.

Comments 4: On page 12, line 259, the authors state, "Upon switching to CO_2 , its polarization curve showed the greatest increase and the current density was higher than that of Vac-air and Ar-air...". However, in Figure 5a, this doesn't seem to align with the description. The highest current density under CO_2 appears to come from the Ar-air sample, and the Vac-air sample doesn't have the highest current difference between Ar and CO_2 . Could the authors clarify this discrepancy?

Response: We are thankful to the reviewer for this valuable suggestion and apologize for the misleading expressions in our previous manuscript. It is important to note that the rotating disc electrode (RDE) measurement system in a CO_2 atmosphere can be approximated to a traditional H-type electrolyzer in CO_2 RR field. In this case, the catalytic interface is dominated by a solid-liquid interface, which means that CO_2 must

first dissolve in the electrolyte and then be converted to a reduced product at the catalytic layer. However, due to the low solubility of CO₂, it is challenging to achieve a CO₂ conversion current density exceeding 30 mA cm⁻² for In-based materials in an H-type electrolyzer (*Angew. Chem. Int. Ed.* **2021**, *60*, 15844; *ChemSusChem* **2020**, *14*, 852). As observed in the oxygen reduction reaction (ORR) study, the current density remains within the range of 1-10 mA cm⁻² even when the rotate speed of the RDE is continuously increased due to the solubility limit of O₂ (*Angew. Chem. Int. Ed.* **2024**, *63*, e202407658; *Adv. Mater.* **2024**, *36*, e2404839). Therefore, in the RDE test conducted in this work, even when Ar was replaced with CO₂, the partial current density of CO₂RR remained below 30 mA cm⁻². The LSV curves obtained with an increasing current density of approximately 200 mA cm⁻² still reflected the electrochemical properties of hydrogen evolution reaction (HER), which were essentially no different from those observed in the LSV curves collected under Ar. It can thus be concluded that the LSV curves in a CO₂ atmosphere, as illustrated in Figure 5a, are not a reliable means of reflecting the difference in CO₂RR catalytic performance among the samples. In light of the distinctive configuration of the flow cell, which can circumvent the current density constraints imposed by CO₂ solubility, we proceeded to record LSV curves for the samples within the flow cell, under both Ar and CO₂ atmospheres. The results were consistent with expectations. As shown in **Figure R2**, the Vac exhibited the minimum current density under Ar, illustrating the most pronounced suppression of HER. Upon switching to CO₂, its polarization curve showed the greatest increase and the current density was higher than that of Vac-air and Ar-air, consistent with its optimum CO₂RR performance.

The variation in current difference between Ar and CO₂ among samples was attributed to the differing concentrations of the HCO₃⁻/CO₃²⁻ buffer pair (*Nat. Commun.* **2024**, *15*, 9145; *Chem. Soc. Rev.*, **2020**, *49*, 6632-6665). As illustrated in Figure 5a, the largest current plateau indicated that when the diffusion of H⁺ reached limit, Ar-air was capable of consuming the local H⁺ in the RDE at a significantly higher rate than Vac-air and Vac, without being replenished by H⁺ in the bulk solution. In this instance, the local pH of Ar-air was higher than that of Vac-air and Vac. Consequently, when the atmosphere was switched to CO₂, Ar-air sample

with a higher OH^- concentration would react with CO_2 to produce sufficient HCO_3^- ($\text{CO}_2 + \text{OH}^- \rightarrow \text{HCO}_3^-$). The buffered HCO_3^- species released H^+ via the conversion between $\text{HCO}_3^-/\text{CO}_3^{2-}$ ($\text{HCO}_3^- \rightarrow \text{CO}_3^{2-} + \text{H}^+$) as the pH continued to rise. It can thus be concluded that the supplementary supply of H^+ in the $\text{HCO}_3^-/\text{CO}_3^{2-}$ buffer pair served to enhance the H^+ reduction current density. The LSV curve revealed that the current difference of Ar-air between Ar and CO_2 was the most pronounced, which could be attributed to the highest concentration of HCO_3^- .

Figure R2. LSV curves of Vac, Vac-air and Ar-air in (a) Ar and (b) CO_2 atmosphere.

Revised parts in the manuscript:

Page 12:

As shown in **Figure 5a**, the Vac exhibited the minimum current density under Ar, illustrating the most pronounced suppression of HER.

Comments 5: On page 13, line 276, the authors mention that j_k and ΔH are plotted in Figure 5c, but they actually appear in Figure 5b. As a result, Figure 5b is not described in the text.

Response: We apologize to the reviewer for this confusing statement. As previously stated in the manuscript, the kinetic-limited current (j_k) and H^+ diffusion coefficient (D_{H^+}) of HER were calculated based on the Koutecký-Levich and Levich equations, respectively. The equations are provided below (for further details, please refer to the Supporting Information):

$$\frac{1}{j_{total}} = \frac{1}{j_K} + \frac{1}{j_{plateau}} = \frac{1}{B} \omega^{-1/2} + \frac{1}{j_K} \quad (\text{Koutecký-Levich equation})$$

$$j_{plateau} = B\omega^{1/2} = 0.62D_{H^+}^{2/3} \nu^{-1/6} nFC^* \omega^{1/2} \quad (\text{Levich equation})$$

Figure 5b illustrated the curve plots of j_{total}^{-1} versus $\omega^{-1/2}$, which was the detailed expression of the Koutecký-Levich equation. The intercept of the curve is j_K^{-1} , in which the j_K could be obtained by multiplicative inverse process. Similarly, **Figure S37** depicted the curve plots of $j_{plateau}$ versus $\omega^{1/2}$, which represented the detailed expression of the Levich equation. The slope of the curve is the proportionality coefficient B , in which the D_{H^+} can be calculated according to the following equation:

$$B = 0.62D_{H^+}^{2/3} \nu^{-1/6} nFAC^*$$

Figure 5c was plotted using the data for j_K and D_{H^+} obtained from **Figure 5b** and **S37**. We hope that our explanation has clarified the reviewer's confusion.

Revised parts in the manuscript:

Page 13:

The kinetic-limited current (j_K) and H^+ diffusion coefficient (D_{H^+}) of HER were calculated according to Koutecký–Levich and Levich equations, respectively (Figure 5b and S29). The results demonstrated that the D_{H^+} of Vac was relatively 10% lower than that of Ar-air, and the corresponding j_K of HER was reduced from 217 mA cm⁻² for Ar-air to 87 mA cm⁻² for Vac (Figure 5c).

Reviewer #4:

The use of a strong acid serves as a means to enhance carbon utilization and mitigate salt precipitation in CO₂ electrolysis. However, challenges such as parasitic hydrogen evolution and electrochemical stability persist in this process. In this study, the authors introduced a new tip-like carbon-coated In₂O₃ catalyst. The carbon layer plays a crucial role in stabilizing the high-valence In³⁺ species, while an electric field induced by the tip structure helps to mitigate the adverse attraction of H⁺ ions and facilitate the desired enrichment of K⁺ ions. This approach was validated through theoretical analysis, a series of spectroscopic examinations, and electrochemical performance evaluations. As a result, they achieved a near-unity FE at 300 mA cm⁻² and demonstrated an initial stability of 100 hours for formic acid production from CO₂ reduction in a strong acid environment. Furthermore, a scaled-up electrolysis employing an active electrode area of up to 25 cm² was successfully conducted at a total current of 7 A.

Overall, this study offers a rational catalyst design strategy to enable efficient CO₂ electrolysis in strong acid conditions. I recommend its publication in this journal following a revision. Detailed comment is provided below:

Response: We are very thankful to the reviewer for the positive opinion and appreciation on our paper. The specific comments have been addressed point-by-point as below. We hope our response and modifications have clarified all the concerns and thereby meet the reviewer's expectations.

Comments 1: The authors stated in their abstract that "Acidic electrochemical CO₂ conversion is a promising alternative to overcome the low CO₂ utilization", yet they did not mention nor discuss this metric in the present work.

Response: We appreciate the reviewer's valuable suggestion. One of the most significant advantages of the acidic CO₂RR system over the alkaline CO₂RR system is that it can markedly enhance the single-pass utilization (SPU) of CO₂ by mitigating the reaction of CO₂ with OH⁻ to form carbonates. We have examined

the SPU of VAC at a current density of 300 mA cm^{-2} with varying CO_2 flow rates. A reduction in the CO_2 flow rate from 20 to 3 standard cubic centimeters per minute (sccm) resulted in an increase in the SPU of CO_2 toward HCOOH , from 11.1% to 70.1% (Figure S43). Our SPU is comparable to the reported values for other acidic systems (*Nat. Commun.* **2024**, *15*, 4821) and is significantly higher than those for alkaline systems, confirming the superiority of acidic CO_2RR systems.

Revised parts in the manuscript:

Page 15:

Furthermore, the single-pass utilization (SPU) of CO_2 towards HCOOH over Vac was assessed at varying CO_2 flow rates. The SPU attained a maximum efficiency of 70.1% at a current density of 300 mA cm^{-2} and a CO_2 flow rate of 3 standard cubic centimeters per minute (sccm) (Figure S43).

Revised parts in the supporting information:

Page S5:

The single-pass utilization (SPU) of CO_2 towards HCOOH was calculated as follow formula at 25°C , 1 atm:

$$\text{SPU} = \frac{j_{\text{total}} \times \text{FE}_{\text{HCOOH}} \times 60}{2 \times F} \div \frac{v \times 1 \times P}{R \times T \times 1000} \times 100\%$$

where the j_{total} represents the total current applied to work electrode, FE_{HCOOH} is the FE for HCOOH .

Page S51:

Figure S43. (a) SPU and (b) HCOOH FE of Vac under different CO_2 gas flow rate at 300 mA cm^{-2} .

Comments 2: This study provides a good demonstration of enhancing CO₂RR in a strong acid environment with 0.1 M K⁺. However, the rationale behind the selection of 0.1 M K⁺ concentration lacks sufficient justification. The authors assert that employing 1 M K⁺ would lead to salt formation and consequently lower stability. This reasoning appears contradictory to their earlier research, where they achieved an extended stability of 5200 hours under similar conditions as reported in *Nature*, 2024, 626, 86-91. Please explain.

Response: We appreciate the reviewer for this instructive comment. It should be noted that the 0.1 M K⁺ concentration was not selected at random. The CO₂RR performance of Vac, Vac-air and Ar-air was evaluated in 0.05 M H₂SO₄ acid electrolyte at 0.02 M, 0.1 M, 0.6 M and 1 M K⁺ concentrations, respectively. The results demonstrated that at a concentration of 1 M K⁺, even the Ar-air sample, which did not exhibit a tip-induced cation enrichment effect, exhibited a similar HCOOH selectivity to Vac due to the excess K⁺. The concentration of K⁺ was reduced to 0.6 M. Despite the slight inadequacy of the K⁺ concentration at this situation, the discrepancy in sample performance was discernible only at a low current density of 50 mA cm⁻². We thus elected to further decrease the K⁺ concentration to 0.1 M. We were gratified to observe that even when the current density was increased from 50 mA cm⁻² to 300 mA cm⁻², the H₂ FE of Ar-air could only be reduced from 56.3% to 13.7%. This contrasted starkly with the 0.6 M K⁺ concentration, where the H₂ FE of Ar-air was suppressed to 10.2% at only 100 mA cm⁻². Meanwhile, Vac benefited from the selective enrichment of K⁺ and the protection of the active In^{δ+} species. At 50 mA cm⁻² in 0.1 M K⁺, its HCOOH FE reaches 80.5%, which was significantly higher than the 56.1% of Vac-air and 43.6% of Ar-air. Upon further reduction of the K⁺ concentration to 0.02 M, the catalytic reaction of Vac catalyst with the lowest K⁺ sensitivity was dominated by HER, reaching a 90.3% FE for H₂ at 50 mA cm⁻², verifying laterally the crucial role of K⁺ for acidic CO₂RR. Therefore, 0.05 M H₂SO₄ with 0.1 M K⁺ was identified as the optimal acidic CO₂RR condition for subsequent characterization analysis.

The exceptional stability observed in our previous work, published in *Nature* (*Nature*, 2024, 626, 86-91), is a

consequence of the strategies we have employed to enhance and repair the gas diffusion electrode (GDE). Specifically, during the preparation of the working electrode, in order to ensure the hydrophobicity of the GDE and to avoid the precipitation of carbonates causing flooding, a 1 wt% PTFE solution was first used to construct a hydrophobic interface on the GDE, after which the catalyst was sprayed. It should be noted that a 1 wt% PTFE dispersion was also added to the catalyst ink during preparation. These precautions ensured that the catalytic reaction could be stable for 200 h and further prolonged the duration time might lead to performance degradation. . Therefore, the extremely long stability of 5200 h was not continuous, and we would stop the electrolysis and repair the working electrode every 200 h. Thereafter, the back of the GDE was coated with a PTFE dispersion for subsequent stability testing until the full 5200-h stability test cycle is complete.

In summary, at a K^+ concentration of 1 M, even with the assistance of GDE enhancement strategies, the single stability test duration could not exceed 200 h. In this work, without any supplementary treatment of the GDE and catalyst ink, the catalytic stability could easily exceed 100 h due to the K^+ concentration of only 0.1 M. This emphasizes the crucial role of reducing the K^+ concentration in the electrolyte to optimize the stability of the catalytic system, which is beneficial for industrial application of CO_2RR technology.

Comments 3: In the introduction section, the necessity of high-valence In species in promoting CO_2RR was not well explained. Instead, they simply stated that “Another factor related to catalytic activity and stability is the anti-reduction property of the catalyst. Performant catalysts with oxidation state suffer from self-reduction under negative bias, inducing electronic structure redistribution of the catalytic sites and consequently compromising the catalytic efficiency”.

Response: We appreciate the reviewer for this professional suggestion. It is generally accepted that metal catalysts in the oxidized state exhibit superior CO_2RR catalytic activity and selectivity compared with their metallic counterparts. The oxygen-bearing Cu catalyst, which exhibited a stable Cu_4O phase, demonstrated a

16-fold enhancement in C₂H₄ selectivity relative to its oxygen-free state (*J. Am. Chem. Soc.* **2020**, *142*, 11417). Similarly, comparable trends were observed in oxide-derived Cu catalysts derived from an oxidation-reduction strategy (*Angew. Chem. Int. Ed.* **2022**, *61*, e202111021). However, it should be noted that electrocatalysts implemented under harsh CO₂RR conditions often exhibit a dynamic nature, which principally causes the evolution of chemical states and accompanying structural reconstruction that can adversely affect the CO₂RR (*Nat. Catal.* **2023**, *6*, 796.). Unfortunately, maintaining the oxidation state of the catalysts during long-term and high-rate operation remains knotty, as the Cu⁺ content can decrease from 100% to 32.1% after 20 min during the CO₂RR (*J. Am. Chem. Soc.* **2020**, *142*, 6400). Similarly, the selectivity and current density of In₂O₃ in CO₂RR-produced formate exhibited a decline over time, which was primarily attributed to the reduction of the catalyst from In₂O₃ to metallic In during electrolysis (*Angew. Chem. Int. Ed.* **2022**, *61*, e202200552). Consequently, it is very necessary to stable the oxidized state of electrocatalysts for the long-term CO₂RR.

Revised parts in the manuscript:

Page 3-4:

Another factor related to catalytic activity and stability is the anti-reduction property of the catalyst. Performant catalysts with oxidation state suffer from self-reduction under negative bias (e.g. the pristine Cu₂O retained only 32.1% of its Cu⁺ component after 20 minutes of CO₂RR electrolysis), inducing electronic structure redistribution of the catalytic sites and consequently compromising the catalytic efficiency. For example, the Faradaic efficiency (FE) of In₂O₃ toward HCOOH was reduced to 70% within 1 h as a consequence of the unavoidable reduction of In₂O₃ to metallic In.

Comments 4: The theoretical computations in this study employed two models of In and In₂O₃ to illustrate the varying catalytic performances from CO₂ to HCOOH. However, in the experimental setup, the authors prepared three catalysts, indicating a discrepancy that requires clarification.

Response: We appreciate the reviewer for this instructive comment. Our DFT calculations focused on

elucidating the role of carbon layer in regulating the properties of In₂O₃. Our calculations demonstrated the coated carbon layer can prevent In₂O₃ from being reduced to metallic In, and In₂O₃ exhibited a much higher CO₂RR activity than In. These results suggested the performance improvement originated from the carbon layer, which was reflected by the differences between the Vac and the Vac-air samples. In addition to the carbon layer, we modified the morphology of the In₂O₃ particles by creating the tip-like structure in experiments. This tip-like structure enhanced the local electric field to enrich K⁺, which was shown in our finite element method simulations and was reflected by the performance differences between the Vac-air and the Ar-air samples. Therefore, our three experimental samples were jointly guided by the DFT calculations and the finite element method simulations.

Comments 5: In Figure 3c-e, the peak at ~1400 cm⁻¹ was assigned to *OCHO species. If so, the related Stark effect should be analyzed. This comment also can be applied to the Raman results in Figure 3f. Alternatively, the peak at ~1400 cm⁻¹ has been previously assigned to bicarbonate in solution (*ACS Catal.* 2020, 10, 8049-8057).

Response: We appreciate the reviewer for this insightful comment. It can be demonstrated that if the carbonate anion (CO₃²⁻) is present in different forms, the assignment of its vibrational peaks will be different. As previously indicated (*ACS Catal.* 2020, 10, 8049-8057), the band center of the solution CO₃²⁻, whether on Au (111) or Cu films, was situated at 1397-1410 cm⁻¹. While, the band center of the adsorbed CO₃²⁻ was located at 1450-1511 cm⁻¹, which was in accordance with our results obtained for the CO₃²⁻ species identified. With regard to the bicarbonate anion, HCO₃⁻, its symmetrical vibration peak was located at 1365 cm⁻¹, which was also in accordance with the results in this work. Therefore, in the manuscript, the peak at 1400 cm⁻¹ could be confirmed as belonging to the *OCHO intermediate, a conclusion that is supported by numerous related reports (*Angew. Chem. Int. Ed.* 2024, 63, e202402070; *Angew. Chem. Int. Ed.* 2024, 63, e202407665).

Furthermore, the Stark effect of the *OCHO intermediate in ATR-SEIRAS and Raman spectra has been

analyzed in accordance with the suggestion of the reviewer. The results were in accordance with the Stark tuning behavior of the $K^+(H_2O)_n$ species as outlined in the original manuscript, where the *OCHO of Vac all exhibited a Stark tuning slope that exceeds that of Vac-air and Ar-air (Figure S39-40). It is important to note that in the Raman spectrum, the stretching vibration peak of the C-H of *OCHO is more prominent than that of O-C-O. Therefore, the C-H vibration at approximately 2850 cm^{-1} was selected as the target for investigating the Stark effect of *OCHO in the Raman spectrum.

Revised parts in the manuscript:

Page 14:

Furthermore, the Stark effect of *OCHO in in situ spectroscopy was analyzed. The results indicated that as the potential became increasingly negative, the frequency shift of *OCHO in Vac also demonstrated the greatest reduction, analogous to that observed in $K^+(H_2O)_n$ (Figure S39-40).

Revised parts in the supporting information:

Page S47-48:

Figure S39. In situ ATR-SEIRAS spectra of (a) Vac, (b) Vac-air, (c) Ar-air. Stark tuning behavior of (d) Vac, (e) Vac-air, (f) Ar-air.

Figure S40. In situ Raman spectra of (a) Vac, (b) Vac-air, (c) Ar-air. Stark tuning behavior of (d) Vac, (e) Vac-air, (f) Ar-air.

Comments 6: In Figure 4, the XRD and XAS results seem to be inconsistent. XRD showed that all samples were mostly reduced to metal-based In with a trace amount of In₂O₃ survived in Vac, whereas XAS results showed the In₂O₃-dominated phase instead of metallic In. Please explain.

Response: We appreciate the reviewer for this professional suggestion. Indeed, the pronounced peaks intensity of the carbon paper and the low crystallinity of the prepared In₂O₃ result in the diffraction peaks of In₂O₃ being markedly weak even in the XRD pattern of the unreacted gas diffusion electrode, as shown in Figure S25. The intensity of the characteristic In₂O₃ peak of the Vac sample after the reaction in Figure 4a is comparable to that observed prior to the reaction (Figure S31), indicating that Vac retains the dominant In₂O₃ component throughout the reaction process. The significantly higher peak intensity of metallic In compared to In₂O₃ can be attributed to the repeated dissolution-redeposition process of dissolved In ions, which facilitates the

continuous growth of reduced metallic In grains while simultaneously enhancing crystallinity. Therefore, the results of XRD and XAS are consistent, both indicating that only trace amounts of the Vac underwent reduction to metallic In during the reaction.

Revised parts in the supporting information:

Page S39:

Figure S31. XRD patterns of Vac gas diffusion electrode before reaction.

Comments 7: Also in Figure 4, the captions of e and f were written in a wrong order.

Response: We are sorry for the mistake in our previous manuscript. In the revised manuscript, the captions order of e and f have been corrected.

Revised parts in the manuscript:

Page 22:

Figure 4. (a) XRD patterns of Vac, Vac-air and Ar-air after CO₂RR. (b) In 3d XPS spectra and (c) O 1s XPS spectra of Vac before and after CO₂RR. (d) In K-edge XANES of Vac before and after CO₂RR. (e) Calculated valence state by linear combination fitting of Vac before and after CO₂RR. (f) FT-EXAFS spectra and (g) WT-EXAFS plots of Vac before and after CO₂RR.

Comments 8: The authors asserted that the Vac sample with carbon-coated In₂O₃ exhibited the highest performance. This raises the question of whether the real active site for CO₂RR is the In atom or the C atom. Additionally, it is important to investigate whether they considered any interface effects on CO₂RR (please refer to Nat. Energy 2020, 5, 478-486)?

Response: We appreciate the reviewer for this professional comment. The carbon-based metal-free electrocatalysts (C-MFECs) are promising options with cost-effectiveness for large-scale electrocatalytic applications. However, for C-MFECs employed in CO₂RR, the products are mainly H₂, as verified by many

studies (*Adv. Energy Mater.* **2017**, *7*, 1700759; *Angew. Chem. Int. Ed.* **2018**, *57*, 13135; *Small Methods* **2021**, *5*, 2001039). The electrocatalytic CO₂RR performance of bare carbon is tested for comparison, which is derived from calcining the ligand (4,5-imidazoledicarboxylic acid and benzimidazole) at 500°C for 1 h under Ar atmosphere. The resulting products are mainly H₂ (**Figure S30**). Therefore, the catalytic performance contribution of carbon in carbon coated In₂O₃ electrocatalyst could be eliminated.

As for the interface effects on CO₂RR, there are many existing studies reporting the carbon-encapsulated metal-based catalysts in CO₂RR. For example, the hydrophobic carbon coated copper core-shell hybrid reported by Yang et al. (*Energy Environ. Sci.* DOI: 10.1039/D1EE01493E) favors the electrosynthesis of methane in CO₂ reduction. And the carbon-encapsulated nickel nanoparticles can selectively reduce CO₂ to CO (*J. Mater. Chem. A*, **2019**, *7*, 6894; *Adv. Energy Mater.* **2019**, *9*, 1902839). In these studies, the selectivity of the hybrid catalysts is generally determined by the employed metal portion, while the carbon encapsulation mainly improves the activity and selectivity of the hybrid composite by altering intermediate adsorption and optimizing electronic structure. In general, the carbon and metal components function in a synergistic way and are taken together contributing to the excellent electrocatalytic CO₂RR performance.

In this work, the interface effects between the carbon layer and the In₂O₃ catalyst were investigated by DFT. The charge density redistribution after carbon confinement demonstrated distinct charge transfer from carbon to In₂O₃. The In₂O₃ without carbon layer protect was easily reduced to metallic In during CO₂RR because of its strong electron affinity. For carbon-coated In₂O₃, the carbon layer served as electron donor to stabilize the In₂O₃ catalyst and maintain the In³⁺ reactive sites. Specifically, the carbon atoms could coordinate with the surface oxygen atoms to increase the formation energy of oxygen vacancies, suppressing the reduction of In₂O₃. More importantly, through DFT, we have confirmed that In³⁺ exhibited a more suitable intermediate adsorption energy barrier than In⁰, and the thermodynamically favorable reaction pathway endowed In₂O₃ with more outstanding CO₂RR catalytic activity. Consequently, as a result of the interaction between the

carbon layer and In_2O_3 , In^{3+} active sites were effectively retained, thereby significantly enhancing the catalytic performance.

The finite element simulation method was employed to elucidate the pivotal function of the carbon layer at the interface of the reaction microenvironment. The tip-like structure led to an enhanced local electronic field because of the high curvature, which is expected to facilitate the adsorption of cations in the electrolyte, i.e., the K^+ and H^+ ions. However, the concomitant attraction of H^+ would break the desired alkaline microenvironment benefited from K^+ enrichment, promoting HER and counteracting the performance improvement from the tip-like structure. Simulation modelling reveals that applying a tightly coated carbon layer on the tip-like catalyst selectively ensures tip-induced K^+ cumulation while eliminating the adverse H^+ enrichment, which is expected to realize efficient acidic CO_2RR at low K^+ concentrations.

In summary, the electrocatalytic CO_2RR performance of the Vac sample outperforms the Vac-air sample and bare carbon, which indicates that the In_2O_3 and carbon layer in Vac function synergistically boosting its excellent electrocatalytic performance. Therefore, it can be concluded that the In_2O_3 is mainly responsible for HCOOH generation, and the carbon layer functions in protecting the oxidation state of In_2O_3 and optimizing local microenvironment, the two portions are taken together guaranteeing the excellent electrocatalytic performance of Vac in acidic environment with low K^+ .

Revised parts in the manuscript:

Page 11:

The ligand-derived bare carbon shown a negligible impact on CO_2RR , which further confirmed that the active site in Vac originated from $\text{In}^{\delta+}$ (Figure S30)

Revised parts in the supporting information:

Page S3:

Preparation of *bare carbon*: The bare carbon was prepared by calcining the ligand (4,5-

imidazolecarboxylic acid and benzimidazole) in an Ar-filled tube furnace at 500 °C for 1 h.

Page S38:

Figure S30. (a) LSV curves and (b) current-dependent H₂ FE of bare carbon in 0.05 M H₂SO₄ with 0.1 M K⁺ under CO₂ atmosphere.

Comments 9: The authors utilized 0.05 M H₂SO₄ as the electrolyte, lacking buffering capacity. Under high current conditions, the electrolyte's pH will notably rise. This issue is particularly critical when employing an H₂SO₄ electrolyte in a 25 cm² electrolytic cell, as maintaining a stable or acidic solution pH becomes challenging. Such pH fluctuations can significantly impact CO₂ utilization. Please explain.

Response: We appreciate the reviewer for this instructive comment. The formation of a dense hydrated cation layer at the outer Helmholtz plane (OHP) using K⁺ ions in an acidic electrolyte to create a local alkaline microenvironment is an important prerequisite for CO₂RR to operate effectively in an acidic environment. Although the local alkaline microenvironment still leads to the formation of (bi)carbonate, the sufficient H⁺ from the acidic bulk solution will promptly neutralize the HCO₃⁻/CO₃²⁻ to form CO₂ for subsequent CO₂RR. Therefore, the utilization rate of CO₂ in acidic CO₂RR systems will be significantly higher than that in alkaline systems. Considering the reviewer's concern that at higher current densities, rapidly consumed H⁺ may lead to an increase in the pH of the acidic bulk solution, which in turn may lead to a decrease in CO₂ utilization. We have investigated the single pass utilization (SPU) in this catalytic system at a current density of 300 mA

cm⁻² with varying CO₂ flow rates. A reduction in the CO₂ flow rate from 20 to 3 standard cubic centimeters per minute (sccm) resulted in an increase in the SPU of CO₂ toward HCOOH, from 11.1% to 70.1% (**Figure S43**). Our SPU is comparable to the reported values for other acidic systems (*Nat. Commun.* **2024**, *15*, 4821) and is significantly higher than those for alkaline systems. This confirmed that operating at the current density in this work did not cause the pH of the acidic electrolyte to rise, thereby attenuating CO₂ SPU. Furthermore, given that the operating current in the 25 cm² scale-up electrolyzer reached the ampere level. In order to minimize the risk of alkalizing the acidic electrolyte and thereby reducing the CO₂ SPU, we have not adopted the traditional feeding method of circulating the electrolyte. As illustrated in **Figure S41**, electrolyte containing 0.05 M H₂SO₄ + 0.05 M K₂SO₄ was continuously fed into the electrolyzer, and the reacted catholyte was directed to an alternative storage tank rather than being returned to the unreacted catholyte tank. This method ensured that the electrolyte in the electrolyzer remained continuously fresh, maintaining a 0.05 M H₂SO₄ concentration and preventing the reduction in CO₂ SPU that would otherwise result from the alkalization of the acidic solution.

Revised parts in the manuscript:

Page 15:

Furthermore, the single-pass utilization (SPU) of CO₂ towards HCOOH over Vac was assessed at varying CO₂ flow rates. The SPU attained a maximum efficiency of 70.1% at a current density of 300 mA cm⁻² and a CO₂ flow rate of 3 standard cubic centimeters per minute (sccm) (Figure S43).

Revised parts in the supporting information:

Page S5:

The single-pass utilization (SPU) of CO₂ towards HCOOH was calculated as follow formula at 25 °C, 1 atm:

$$\text{SPU} = \frac{j_{\text{total}} \times \text{FE}_{\text{HCOOH}} \times 60}{2 \times F} \div \frac{v \times 1 \times P}{R \times T \times 1000} \times 100\%$$

where the j_{total} represents the total current applied to work electrode, FE_{HCOOH} is the FE for HCOOH.

Figure S43. (a) SPU and (b) HCOOH FE of Vac under different CO₂ gas flow rate at 300mA cm⁻².

Figure S41. Photographs of the two-electrode scale-up electrolyzer configuration.

Finally, we again appreciate the reviewers for your valuable time and insightful comments, which were extremely helpful in improving the quality of our paper and enhancing our future work. We sincerely hope our responses and revised manuscript have clarified the reviewer's concerns and the reviewer could recognize and agree with our efforts, and the revised manuscript will be deemed acceptable for publication in *Nature Communications*.

Dear Reviewers,

Many thanks for your valuable time and work on our manuscript (**Manuscript ID: NCOMMS-24-57974B**). We have revised the manuscript carefully by supplementing additional revisions and discussions. We sincerely appreciate your comments, which have greatly improved our manuscript. Details about the revisions and our responses to the reviewers' comments are provided below in our point-by-point response.

Reviewers' comments:

Reviewer #1:

The authors have carefully addressed my concerns, so it is now recommended.

Response: We thank the reviewer for the strong support in the publication of this work.

Reviewer #2:

I have re-read the paper. The authors have revised their manuscript.

Response: We thank the reviewer very much for the positive comment.

Reviewer #3:

The authors replied and revised all my questions. I recommend it to be published.

Response: We thank the reviewer for the positive recommendation for publication.

Reviewer #4:

The authors have taken into consideration all of my feedback. Therefore, I highly recommend its publication with just one minor suggestion. I suggest that the authors revisit the fitting baselines in Figs. 4b and 4c. Specifically, the XPS fitting baseline should be a smooth line rather than a fluctuating one.

Response: We thank the reviewer for the valuable comments and positive recommendation for publication.

In the revised manuscript, we have corrected the fitting baselines in Figure 4b and 4c.

Fig. 4 Characterization of catalysts after CO₂RR. a XRD patterns of Vac, Vac-air and Ar-air after CO₂RR. **b** In 3d XPS spectra and **(c)** O 1s XPS spectra of Vac before and after CO₂RR. **d** In K-edge XANES of Vac before and after CO₂RR. **e** Calculated valance state by linear combination fitting of Vac before and after CO₂RR. **f** FT-EXAFS spectra and **(g)** WT-EXAFS plots of Vac before and after CO₂RR.

Finally, we again appreciate the reviewers for your valuable time and insightful comments, which were extremely helpful in improving the quality of our paper and enhancing our future work.